

# Rainfall retrieval with commercial microwave links in São Paulo, Brazil

Manuel F. Rios Gaona[1,4], Aart Overeem[1,2], Timothy H. Raupach[3], Hidde Leijnse[2], and Remko Uijlenhoet[1]

[1]Hydrology and Quantitative Water Management Group, Department of Environmental Sciences, Wageningen University, 6708 PB Wageningen, the Netherlands.
[2]R&D Observations and Data Technology, Royal Netherlands Meteorological Institute, 3731 GA De Bilt, the Netherlands.
[3]Environmental Remote Sensing Laboratory, École Polytechnique Fédérale de Lausanne, Lausanne, Switzerland.
[4]IIHR - Hydroscience & Engineering, The University of Iowa, Stanley Hydraulics Lab (SHL), Iowa City, USA.

*Correspondence to:* Manuel F. Rios Gaona (manuelfelipe-riosgaona@uiowa.edu)

**Abstract.** In the last decade there has been a growing interest from the hydrometeorological community regarding rainfall estimation from commercial microwave link (CML) networks. Path-averaged rainfall intensities can be retrieved from the signal attenuation between cell phone towers. Although this technique is still in development, it offers great opportunities for the retrieval of rainfall rates at high spatiotemporal resolutions very close to the Earth's surface. Rainfall measurements at high

5  spatiotemporal resolutions are highly valued in urban hydrology, for instance, given the large impact that flash floods exert on society. Flash floods are triggered by intense rainfall events that develop over short time scales.

Here, we present one of the first evaluations of this measurement technique for a subtropical climate. Rainfall estimation for subtropical climates is highly relevant, since many countries with few surface rainfall observations are located in such areas. The test bed of the current study is the Brazilian city of São Paulo. The open-source algorithm RAINLINK was applied to

10  retrieve rainfall intensities from (power) attenuation measurements. The performance of RAINLINK estimates was evaluated for 5 of the 250 CML in the São Paulo metropolitan area for which we received data, for 81 days between October 2014 and January 2015. We evaluated the retrieved rainfall intensities and accumulations from CML against those from a dense automatic gauge network. Results were found to be promising and encouraging, especially for short links, for which high correlations ($> 0.9$) and low biases ($\sim 30\%$ and lower) were obtained.

## 1 Introduction

Rainfall is the key input in environmental applications such as hydrological modeling, flash-flood and crop growth forecasting, landslide triggering, quantification of fresh water availability, and waterborne disease propagation. Because it is a natural process with a high spatiotemporal variability (Hou et al., 2008; Sene, 2013b), its accurate estimation is a demanding task.

The most common technologies that are currently used to measure rainfall at larger scales are rain gauges, radars and satelli-

20  tes. Each technology presents advantages and drawbacks with regard to the accuracy of rainfall estimates and the spatiotemporal coverage. Rain gauges directly measure the quantity of precipitation that falls on the ground. They offer accurate estimates of



rainfall collected at temporal scales from minutes to days. Nevertheless, their rainfall estimates are only representative of their direct vicinity. In addition, in most cases the gauges within a network are unevenly distributed in space. Weather radar (RAdio Detection And Ranging) offers indirect estimates of rainfall, with horizontal resolutions of $\sim$1 km (or even less depending on the radar settings) every $\sim$2 to $\sim$5 min. They scan distances of $\sim$100$-$300 km, which represent an area of $\sim$125,000 km$^2$, if

issues of beam blockage are not present. The accuracy of rainfall estimates from radar depends on how well the measurements of backscattered power from hydrometeors are transformed into rain rates. Satellites offer also indirect estimates of rainfall at several spatiotemporal resolutions. For instance, Geostationary Earth Orbit (GEO) satellites (orbiting the Earth at $\sim$36,000 km) provide observations at resolutions of $\sim$10$-$60 min, and 1$-$4 km (Sene, 2013a; Wang, 2013), whereas Low Earth Orbit (LEO) satellites (orbiting the Earth at $\sim$800 km) can provide observations at resolutions of $\sim$1 km or less. Gridded rainfall products

from the Global Precipitation Measurement mission (GPM) offer precipitation estimates between 60°N$-$60°S at a spatial resolution of 0.1°$\times$0.1° every 30 min. The main advantage of satellites above radars and gauges is that they provide global rainfall estimates (oceans included).

Commercial microwave links (CML) represent a technology that in the past decade has gained momentum as an alternative means for rainfall estimation. CML rainfall estimates are more representative of rainfall at the ground surface than those offered

by satellites and/or weather radars. Networks of CML are more dense than gauge networks given their worldwide deployment for telecommunication purposes (Overeem et al., 2016b; Kidd et al., 2017). This worldwide spread of CML potentially offers rainfall estimates in places where rain gauges are scarce or poorly maintained, or where ground-based weather radars are not yet deployed or cannot be afforded. The spatiotemporal resolution of rainfall estimates from CML varies from seconds to minutes, and from hundreds of meters to tens of kilometers. For instance, Messer et al. (2012), and Overeem et al. (2016b) use

maximum and minimum Received Signal Level (RSL) measurements over 15-min intervals, for CML with (spatial) densities of 0.3 to 3 links per km$^2$, and 0.1 to 2.1 km per km$^2$, respectively. Fencl et al. (2015), and Messer et al. (2012) provide 1-min rainfall estimates, whereas 1-s retrievals are obtained by Chwala et al. (2016), and Doumounia et al. (2014).

The interaction between attenuation and rainfall has long been studied by the electrical engineering community (from the attenuation perspective), and since the last two and a half decades by the hydrological community (from the rainfall perspective).

Hogg (1968) and Crane (1971) review the influence of atmospheric phenomena on mm- and cm-wavelength based satellite communication systems. Later, Hogg and Chu (1975) and Crane (1977) focus exclusively on the role of rainfall in satellite communication, as rainfall is the major source of propagation issues for frequencies above 4$-$10 GHz. Recently, Badron et al. (2011) and Chakravarty and Maitra (2010) study rain-induced attenuation in satellite communication at tropical locations, where the attenuation is severe. Even more recently, Barthès and Mallet (2013) and Mercier et al. (2015) retrieve high reso-

lution rainfall fields (0.5$\times$0.5 km every 10 sec) from 10.7- and 12.7-GHz Earth-space links used in satellite TV transmission, even though at Ku band the estimation of weak rainfall rates is not optimal.

Our main interest here is rainfall estimation from terrestrial links. The idea of rain rate retrieval from attenuation measurements via tomographic techniques was presented by Giuli et al. (1991). Cuccoli et al. (2013) and D'Amico et al. (2016) present reconstructed 2D-rainfall fields from operational ML networks via tomographic techniques. Ruf et al. (1996) use a 35-GHz

dual-polarization link for rainfall estimation at 0.1$-$1 km horizontal resolutions. Holt et al. (2000), Rahimi et al. (2004) and



Upton et al. (2005) estimate path-averaged rainfall from the differential attenuation of dual-frequency links. Minda and Naka-mura (2005) use a 50-GHz link of 820 m to estimate rainfall. At such frequencies (or higher) rainfall estimation is sensitive to the raindrop size distribution and raindrop temperature. The synergistic use of ML, gauges and radars for rainfall estimation is proposed by Grum et al. (2005) and Bianchi et al. (2013). The first references to rainfall estimates from CML are Messer et al.

(2006) and Leijnse et al. (2007). Berne and Uijlenhoet (2007), Leijnse et al. (2010), and Zinevich et al. (2010) study sources of uncertainty in rainfall estimates from CML. Methods for country-wide rainfall fields from CML are developed in Zinevich et al. (2008) and Overeem et al. (2013).

In the last decade the use of CML has broadened its spectrum to several other environmental applications beyond rainfall estimation, for instance, melting snow (Upton et al., 2007), water vapour monitoring (David et al., 2009), wind velocity esti-

mation (Messer et al., 2012), dense-fog monitoring (David et al., 2013), urban drainage modelling (Fencl et al., 2013), flash flood early warning in Africa (Hoedjes et al., 2014), and air pollution detection (David and Gao, 2016).

We evaluate the performance of 5 CML located in the city of São Paulo, Brazil, in terms of their capacity to retrieve rainfall for the period between 20 October 2014 and 9 January 2015 ($\sim$3 months). Rainfall evaluation against gauge data was coherently possible for 5 links from a network of 250 CML. Previously, da Silva Mello et al. (2002) studied the attenuation along ML due

to rainfall for São Paulo. They used 6 links ($7-43$ km) with frequencies between 15 and 18 GHz. Here, instead of considering rainfall to be a nuisance for the propagation of radio signals, we invert the problem by considering the attenuation suffered by such signals to be a valuable source of rainfall information. Since CML were not intended for rainfall estimation purposes, these devices can be considered a form of opportunistic sensors. They are potentially cost-free as the retrieved rain rates can be regarded as a by-product of power measurements.

As subtropical and tropical regions are the ones most deprived of radar (Heistermann et al., 2013) and gauge networks (Kidd et al., 2017; Lorenz and Kunstmann, 2012), CML could serve as complementary (or even alternative) networks for rainfall monitoring. Most of the recent studies concerning rainfall retrieval from CML have focused on temperate and Mediterranean climates, e.g., Overeem et al. (2016b); Messer and Sendik (2015). Thus, our evaluation is one of the first which focuses on a subtropical climate, complementing the study of Doumounia et al. (2014), which focused on a semi-arid climate. Focus on

accurate rainfall estimation within the subtropics is of high relevance given that in such regions (e.g., São Paulo) intense events develop more often into flash floods and mud slides, which cause damage to property, disruption of business, and occasional casualties (Pereira Filho, 2012).

This paper is organized as follows: Section 2 describes the study area, the datasets (CML, rain gauges, disdrometers), the retrieval algorithm, and the evaluation metrics. The results and related discussion of our major findings are presented alongside

in section 3. Summary, conclusions and recommendations are provided in section 4.



## 2 Study Area, Data and Methods

### 2.1 Description of Study Area

The city of São Paulo is located $\sim 60\,\text{km}$ from the Atlantic Ocean at $\sim 770\,\text{masl}$, where sea breeze fronts commonly push from the SE against prevailing continental NW winds (cold fronts). In general, the incoming sea breeze interacts with the warmer and drier (urban) heat island of São Paulo, producing very deep convection with heavy rainfall, wind gusts, lightning and hail (Vemado and Pereira Filho, 2016; Machado et al., 2014; Pereira Filho, 2012). de Oliveira et al. (2002) characterize the local climate as typical of subtropical regions of Brazil, with a dry winter (June-August) and a wet summer (December-March). With regard to the climatology of São Paulo[1], February is the warmest month with $22.4°\,\text{C}$, and July the coldest with $15.8°\,\text{C}$. Climatological averages for temperature and humidity, for November and December (the full two months of the studied period), are 20.2 and $21.1°\text{C}$, and $78\%$ and $80\%$, respectively. August is the driest month with $39.6\,\text{mm}$ of precipitation, and January the wettest with $237.4\,\text{mm}$, on average. The (climatological) yearly accumulated rainfall is $1441\,\text{mm}$. Overeem et al. (2016b) report winter time issues in rainfall estimates from CML, i.e. solid and melting precipitation. However, for the subtropical climate of São Paulo, such winter issues are not expected to play a role, which is advantageous for accurate rainfall estimation.

### 2.2 Data

We received power measurements from two brands of CML: Ericsson (ER) and Huawei (HU). Power levels were registered every 15 min from 0100 UTC 20 October 2014 to 0045 UTC 8 January 2015, i.e., 81 days exactly. Their quantization level was 0.1 dB. Minimum and maximum levels of received and transmitted power were available for 101 HU CML (Fig. 1), whereas only minimum received powers were available for ER CML (149). Because our CML retrieval algorithm RAINLINK (Sec. 2.3) only retrieves rain rates from minimum and maximum power levels, we discarded the ER CML. Due to issues in the log-file of the attenuation measurements, it was only possible to correctly and unequivocally assign power levels to 66 HU CML (16 full-duplex and 34 simplex). From the 66 HU CML, we selected 17 CML given their proximity to rain gauges (1 km or less). Our experience tells us that CML with both lengths above 20 km and frequencies above 15 GHz are not common in CML networks (they are highly unlikely from a network design perspective: long links experience more attenuation in rain, and should hence operate at low frequencies to limit this attenuation). Hence, we discarded 6 CML as dubious and did not consider them in our analyses, which reduced the number of CML to 11. Finally, from the 11 remaining CML, we only kept the 5 CML which showed clear rainfall signals as compared to nearby rain gauges, i.e. for which $r^2 \geqslant 0.7$. The other 6 CML practically showed no correlation with nearby gauges ($r^2 \sim 0.3$ for one CML-gauge pair, and $r^2 < 0.1$ for the other 5 CML-gauge pairs), due to malfunctioning gauges and/or CML data issues. Figure 2 shows the scatter plot of frequency against length for all HU CML. For RAINLINK to work, it is necessary that the power level of the transmitted signal is essentially constant. For the

---

[1]The climatological data presented here cover the period from 1961 to 1990 and correspond to the station "Mir. de Santana" located in the heart of São Paulo city ($-46.6\,\text{lon}$, $-23.5\,\text{lat}$, $792\,\text{masl}$). These data are freely available at the INMET (METeorological National Institute) web portal: http://www.inmet. gov.br/portal/index.php?r=home2/index.





remaining 5 CML evaluated here, the mean difference between 15-min transmitted power levels is ∼0.0 dB, with a maximum of 0.5 dB (for the 81 days considered).

Rainfall depths from 152 stations were retrieved from the National Early Warning and Monitoring Centre of Natural Disasters (CEMADEN), Brazil[2]. These 152 stations offer 10-min rainfall depths for the period and region under study (Fig. 1).

Stations located within 1 km distance from the evaluated link paths were selected. Hence, only 11 stations were used to evaluate CML rainfall estimates in São Paulo.

Thanks to the CHUVA project (Machado et al., 2014), we retrieved 1-min drop size distributions (DSD) from 3 first-generation Parsivel disdrometers located in the region "Vale do Pariba", ∼93 km east of the study area[3]. These DSD data were collected from 1 November 2011 to 14 March 2012. The DSD recorded by the Parsivels were corrected by the method of

Raupach and Berne (2015a, b). We use updated correction factors trained from French disdrometer data. Due to instrumental, climatic, and location differences, these correction factors are taken as approximations.

## 2.3 Rainfall Retrieval Algorithm

Rainfall estimation from CML is based on power measurements from the electromagnetic signal along a link path, i.e., between transmitter and receiver. Rainfall rates can be retrieved from the decrease in power, which is largely due to the attenuation of

the electromagnetic signal by raindrops along the link path. The power-law relation between attenuation and rainfall (along a link path) was established by Olsen et al. (1978) and Atlas and Ulbrich (1977) as:

$$k = aR^b, \tag{1}$$

where $k$ is the specific attenuation [dB·km$^{-1}$] along the link path attributed to rainfall and $R$ is the rainfall rate [mm·h$^{-1}$]. The coefficient $a$ and exponent $b$ depend on the frequency and polarization of the electromagnetic signal, the DSD, and (to a

much lesser extent) on the raindrop temperature. In the frequencies at which CML commonly operate, the exponent $b$ in Eq. (1) is ∼1.0. Atlas and Ulbrich (1977) state that the near-linearity between rain rates and specific attenuation (in the 20−40 GHz band) "makes it possible to use the total path loss as a direct measure of $\overline{R}$ [average rain rate] independent of the form of the distribution of $R$ [rain rate] along the path".

Both the degree to which Eq. (1) holds and the values of $a$ and $b$ are determined by the DSD. In order to study how strongly

this relation deviates from other relations found in the literature, we determine values of $a$ and $b$ based on measured DSDs from the São Paulo region. For each 1-min DSD, we compute the corresponding rainfall intensity and specific attenuation at the 3 most common frequencies in São Paulo (11, 18, and 23 GHz). Specific attenuation is computed for vertically polarized signals (most CML operate using this polarization) using T-Matrix scattering computations (e.g. Mishchenko, 2000), assuming

---

[2]Gauge data from Brazil is freely available at http://www.cemaden.gov.br/mapainterativo/.

[3]DSD data from Parsivel and other disdrometers for the region of São Paulo (and other regions across Brazil) are freely available at http://chuvaproject.cptec.inpe.br/soschuva/

[4]We received CML data from a third party. It was not possible to verify the topology of the network, shown in Fig. 1 on-site, which we suspect not always to be accurate given the orientation of the long links.





raindrop oblateness as a function of its volume-equivalent diameter given by Andsager et al. (1999), and an average raindrop temperature of $298.36\,\mathrm{K}$. The values of $a$ and $b$ are subsequently determined in log–log space (orthogonal regression) by a linear fit of $R = ak^b$ to the computed values of $R$ and $k$. Note that conversely to Eq. (1) $\log(R)$ is the dependent variable in this case.

Figure 3 shows the power-law relations for the 3 frequencies. This figure also shows the power-law relations derived for rainfall in the Netherlands (Leijnse, 2007, p. 65), and those recommended by the International Telecommunication Union (ITU-R Recommendation P.838-3). It is clear from this figure that there certainly are differences, and that such differences are largest for $11\,\mathrm{GHz}$ at high rainfall intensities. For the higher frequencies, such differences are more limited, especially at high rainfall intensities. This is in line with what has been found earlier (e.g. Berne and Uijlenhoet, 2007; Leijnse et al., 2008, 2010).

RAINLINK (Overeem et al., 2016a) is an R package (R Core Team, 2016) in which rain rates and area-wide rainfall maps can be derived from CML attenuation measurements. A very brief description of the algorithm is as follows: 1) wet-dry classification – a link is considered for non-zero rainfall retrievals if the received power jointly decreases with that of nearby links ($50\,\mathrm{km}$ radius for this study); 2) reference signal level estimation – the median signal level of all dry periods in the previous $24\,\mathrm{h}$ is considered as the representative level of dry weather; 3) outlier removal – exclusion of links for which the specific attenuation (accumulated over $24\,\mathrm{h}$) deviates too much from that of nearby links; 4) rainfall retrievals – once attenuation estimates are obtained from the difference between RSL and the reference signal level, 15-min average rainfall intensities are computed from a weighted average of minimum and maximum rainfall intensities obtained by the (inverse) power-law of Eq. (1); and 5) rainfall maps – rainfall intensities are interpolated into rainfall maps through Ordinary Kriging. This latter step was not implemented in this study. Overeem et al. (2016a) give a more detailed and in-depth review and description about all the technicalities within the RAINLINK package.

## 2.4 Error and Uncertainty Metrics

We evaluated the rainfall estimates from RAINLINK through: 1) the relative bias, 2) the coefficient of variation (CV), and 3) the coefficient of determination ($r^2$).

For a given CML, the relative bias is a relative measure of the average error between the RAINLINK estimates $R_{\mathrm{RAINLINK},i}$ and the rain gauge measurements $R_{\mathrm{gauge},i}$ (the latter considered as the ground truth):

$$\text{relative bias} = \frac{\overline{R}_{\mathrm{res}}}{\overline{R}_{\mathrm{gauge}}} = \frac{\sum\limits_{i=1}^{n} R_{\mathrm{res},i}}{\sum\limits_{i=1}^{n} R_{\mathrm{gauge},i}}, \tag{2}$$

where $R_{\mathrm{res},i} = R_{\mathrm{RAINLINK},i} - R_{\mathrm{gauge},i}$ and $n$ represents all possible time steps for the (rainfall) event under consideration. $R_{\mathrm{res},i}$ are the residuals, i.e., the difference between $R_{\mathrm{RAINLINK},i}$ and $R_{\mathrm{gauge},i}$. $\overline{R}_{\mathrm{res}}$ and $\overline{R}_{\mathrm{gauge}}$ are the average of the residuals and gauge rainfall measurements (in mm), respectively. The relative bias ranges from $-1$ to $+\infty$, where 0 represents unbiased rainfall estimates.





The coefficient of variation is a dimensionless measure of dispersion (Haan, 1977), defined in this case as the standard deviation of the residuals $\sqrt{\widehat{\mathrm{Var}}\left(R_\mathrm{res}\right)}$ divided by the mean of the rain gauge measurements, for the evaluated CML:

$$\mathrm{CV} = \frac{\sqrt{\widehat{\mathrm{Var}}\left(R_\mathrm{res}\right)}}{\overline{R}_\mathrm{gauge}}. \tag{3}$$

The CV is a measure of uncertainty. It ranges from $0$ (a hypothetical case with no uncertainty) to $\infty$.

The coefficient of determination is a measure of the strength of the linear dependence between two random variables, RAINLINK estimates and rain gauge measurements in this case. It is defined as the square of the correlation coefficient between $R_{\mathrm{RAINLINK},i}$ and $R_{\mathrm{gauge},i}$:

$$r^2 = \frac{\widehat{\mathrm{Cov}}^2\left(R_\mathrm{gauge}, R_\mathrm{RAINLINK}\right)}{\widehat{\mathrm{Var}}\left(R_\mathrm{gauge}\right) \cdot \widehat{\mathrm{Var}}\left(R_\mathrm{RAINLINK}\right)}, \tag{4}$$

where $\widehat{\mathrm{Var}}\left(R_\mathrm{gauge}\right)$ and $\widehat{\mathrm{Var}}\left(R_\mathrm{RAINLINK}\right)$ are the variance of rain gauge measurements and RAINLINK estimates, respectively; and $\widehat{\mathrm{Cov}}^2\left(R_\mathrm{gauge}, R_\mathrm{RAINLINK}\right)$ the squared covariance between these two variables. $r^2$ ranges from $0$ to $1$, this latter the case of perfect linear correlation, i.e., all data points would fall on a straight line without any scatter. Perfect linearity does not imply unbiased estimates because the regression line does not have to coincide with the 1:1 line, even if it captures all variability.

The metrics were systematically computed on 30-min paired rainfall depths, both above $0\,\mathrm{mm}$ (to only account for significant rainfall events), and for which their equivalent 15-min minimum received powers (i.e., "min PRx ..." in Fig. 4) were larger than $-90\,\mathrm{dB}$. 30-min aggregation was necessary given the temporal resolutions of the datasets, i.e., $10\,\mathrm{min}$ for gauge and $15\,\mathrm{min}$ for link-retrieved data.

## 3   Results and Discussion

### 3.1   Evaluation of 30-min Rainfall

Figure 4 shows minimum and maximum received powers and the derived CML rainfall rates at 15-min resolution, as well as the rain rates from the nearest gauge at 10-min resolution. The upscaled 30-min rainfall rates from both CML and gauges are also shown in Fig. 4. It can be seen that the minimum and maximum received powers are strongly negatively correlated with the gauge rainfall rates. The figure presents the two longest rainfall events for CML 14. One can see that the stronger the rainfall event is, the larger is the attenuation registered by this CML.

Uncertainties in gauge and attenuation measurements themselves are the two sources of error that mainly constrain our evaluation. Our work compares CML rainfall estimates against rain gauge measurements, which are considered here as the "ground truth". Nonetheless, a gauge is only representative of its surrounding area and does not account for the spatial variability of rainfall along the link path. Representativeness errors will increase for longer link paths and for more intense rainfall events. For subtropical regions where intense rainfall is associated with small convective raincells, da Silva Mello et al. (2014) showed that due to smaller raincells only a part of the link-path contributes to the attenuation, which causes an effective link-rain rate smaller than the one(s) measured by gauges.




The results of Fig. 4 are obtained for the longest link (5.3 km), where representativeness errors could play a larger role. The worst results are found for the shortest links ($< 1.0$ km). This may be related to the fact that rain-induced attenuation along the link path may be relatively small compared to the attenuation caused by wet antennas, i.e., the wet antennas may explain the large overestimation found for CML 13 and 12 (see Tables 1 and 2).

Table 1 provides an extensive verification of 5 of the CML shown in Fig. 1. The results presented in Table 1 (and in Table 2) come from the automatic selection of significant rainfall events for each particular CML. A rainy event is identified if both rainfall depths (CML and gauges) are above $0.0$ mm for one or more consecutive 30-min steps, i.e., if it lasted 30 min or more. A 30-min step is only considered if the received powers from which CML rainfall was retrieved were larger than $-90$ dB. Hence, if the evaluation of CML estimates yields a $r^2 \geqslant 0.7$ we can be sure that CML and gauge estimates show similarities, which is the case for the selected 5 CML in Tables 1 and 2. Given the fact that the CML and gauge measurements are totally independent, it is likely that the high values of $r^2$ indicate that both types of observations contain a true rain signal.

A high value of $r^2$, although promising, does not automatically imply that CML rainfall estimates are accurate. Hence, it is important to also compute other metrics, such as CV and the relative bias. CML 06, 07 and 14 provide relatively low values of CV, i.e. below 1. The relative bias for CML 06 and 07 is relatively high ($\sim 15-32\%$), whereas the relative bias for CML 14 is within $16\%$. CML 12 and 13 do likely perceive a rain signal given the high values of $r^2$, but their values of CV are much larger than 1, and their relative bias is roughly $125-152\%$, implying a large overestimation of rainfall by CML. This might be caused by wet antenna attenuation being more dominant for those short links, although the overestimation is not as strong as for the short CML 06. Alternatively, this high correlation and large bias might be caused as well by errors in the CML metadata, i.e, wrong location of one of the antennas or wrong frequency. A similar behaviour was found in some of the CML we discarded.

The results for different $R-k$ relations are quite similar, indicating that differences in DSD climatologies play a smaller role. For CML 12 and 13 the relative bias becomes less severe for the $R-k$ relation derived from São Paulo data. In general, local parameters (i.e., SP) are the best approach for RAINLINK. Nevertheless, the RAINLINK default parameters offer Brazilian (subtropical) CML estimates of reasonable quality despite the local (temperate) climate for which they were obtained, namely the Netherlands. The ITU parameters often lead to a much higher value of CV, and always to a larger overestimation.

## 3.2 Evaluation of Event Rainfall Accumulations

We further explore the performance of these 5 CML by studying all registered rainfall events for the period under consideration, i.e, $\sim 60$ rainy events (23 for CML 06 due to data availability issues; Table 2). Figure 5 presents a scatter plot of aggregated CML rainfall against aggregated gauge rainfall for all the 272 rainfall events summarized in Table 2 per CML. In Fig. 5 one can see how CML 07 and 14 accurately measure very intense, as well as light rainfall events. CML 14 is the one with less variability than CML 07, and the one which outperforms all other four CML.

The clear potential of CML technology for rainfall estimation (for subtropical climates) is presented through the outstanding performance of CML 14 (Fig. 4, and Tables 1 and 2). On average, and for the aggregated rainfall of the 65 events of CML 14, the $r^2$ is 0.91, with a very low CV of 0.5, and a relative bias of only $10\%$ (Table 2). Hence, the rainfall estimates from CML 14 agree well with those from a gauge based on a large dataset. This gives an indication that the RAINLINK package is suitable



to retrieve rainfall via CML data from a subtropical climate, even though many of its parameters have not been optimized for such a climate. Since biases propagate in hydrological model forecasts, the low relative bias found for CML 14 is important if its rainfall estimates would be used as input in a hydrological model.

The results presented in Fig. 5 correspond to RAINLINK retrievals where no wet-dry classification is applied, in order to focus on the performance of individual CML without information from surrounding CML. Moreover, this shows the performance of RAINLINK in case few nearby CML are available. If the wet-dry classification from RAINLINK is applied, i.e. the nearby link approach, slightly improved metrics are obtained (not shown here).

## 4    Summary, Conclusions and Recommendations

CML networks are an opportunistic technique for rainfall estimation, with the potential to be used worldwide given the spread of CML-based telecommunication systems during the last two decades. Here we presented one of the first evaluations of CML rainfall retrievals for a subtropical climate. Subtropical regions could benefit from this technique given that rainfall events are often more extreme, and usually fewer surface rainfall measurements are collected. We evaluated rainfall retrievals from power measurements for 5 CML from a network located in the city of São Paulo. We used RAINLINK (Overeem et al., 2016a) to retrieve rainfall from these CML.

30-min rainfall estimates from CML were evaluated against rainfall measurements from the nearest rain gauge for the period from 20 October 2014 to 9 January 2015. We focused our analyses on the 5 CML for which $r^2 \geqslant 0.7$. Three out of five CML gave good results in terms of CV. One CML also had a low relative bias. Subsequently, the quality of rainfall estimates from these 5 CML was also evaluated in terms of cumulative rainfall from 272 events. The good results indicate that RAINLINK can successfully be applied to CML data from a subtropical climate, even though most parameters have been optimized for the temperate climate of the Netherlands.

In order for RAINLINK to capture the rainfall characteristics from the region of São Paulo, we derived $a$-$b$ coefficients of power-law $R-k$ relations from local DSD data. The $a$ and $b$ coefficients are a function of the polarization and frequency of the link, DSD and raindrop temperature. These local DSD parameters gave the best results, whereas the ITU/NL parameters proved to be very useful and accurate enough when local $a$-$b$ coefficients cannot be derived. The NL parameters are characteristic for the hydroclimatology of the Netherlands, and are the default set in RAINLINK. They also outperform the ITU parameters.

The 5.3-km CML was the one with the best performance for the SP $R-k$ relation, i.e., $r^2 = 0.735$, CV $= 0.72$, and relative bias of $-7.1\%$. Such a low relative bias indicates the suitability of CML rainfall retrievals in hydrological modeling, for instance.

A more thorough evaluation should be done to study and explain differences between CML and gauge rainfall estimates. For instance, the influence of rainfall variability along link paths could be studied. This can be achieved if local radar measurements are compared against CML estimates, which would allow to better track the rain events and their incidence over the link paths, especially relevant for longer link paths. We also encourage future work on sensitivity analysis focused on the optimization of RAINLINK parameters to improve the accuracy of rainfall estimates in subtropical regions. Missing maximum signal level




data, and unexpected combinations of link lengths and microwave frequencies, forced us to remove many CML from the original dataset. This shows that accurate metadata, such as link coordinates for instance, are essential.

CML are not the replacement of current standard technologies such as radars, rain gauges (and even satellites), but their opportunistic use is rather valuable as complementary networks for high-resolution rainfall estimation. To conclude, we were able to obtain good results for one link, which confirms the great potential of this technique if the data and metadata are properly stored.

*Author contributions.* M.F. Rios Gaona sorted, analysed and plotted the data, and wrote most of the paper. T. Raupach processed the Parsivel DSD data. A. Overeem, H. Leijnse, and R. Uijlenhoet analysed the results, gave valuable feedback, and wrote parts of the paper.

*Competing interests.* The authors manifest not to have competing interests with the Planetary Skin Institute / Italia Mobile, which provided us the CML data.

*Acknowledgements.* We thank Juan Carlos Castilla-Rubio (from Planetary Skin Institute); the Brazilian Ministry of Science, Technology and Innovation; and Italia Mobile for providing the CML data. This work was financially supported by The Netherlands Organisation for Scientific Research NWO (project ALW-GO-AO/11-15).



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



**Figure 1.** Topology of one CML network in the city of São Paulo, Brazil. 101 Huawei (HU) CML are shown. CML in red are the ones where unequivocal power level assignment was feasible (HU_data). CML which have both lengths above 20 km and frequencies above 15 GHz were not analyzed. Thus, only 11 CML were retained given the proximity of rain gauges to their link paths, i.e., $\leqslant 1$ km (yellow circles). All circles represent gauges with 10-min resolution available for the studied period (20 October 2014 to 9 January 2015). The black numbers refer to the CML that showed clear rainfall signals as compared to nearby gauges, i.e. for which $r^2 \geqslant 0.7$ (see also Table 1). The letters refer to the corresponding gauges. CML data was provided by the Planetary Skin Institute / Italia Mobile[4]. Gauge data was retrieved from the CEMADEN database. The geographical location of São Paulo is given in the upper left corner. The DEM was extracted from Google Maps (Google Maps, 2017).



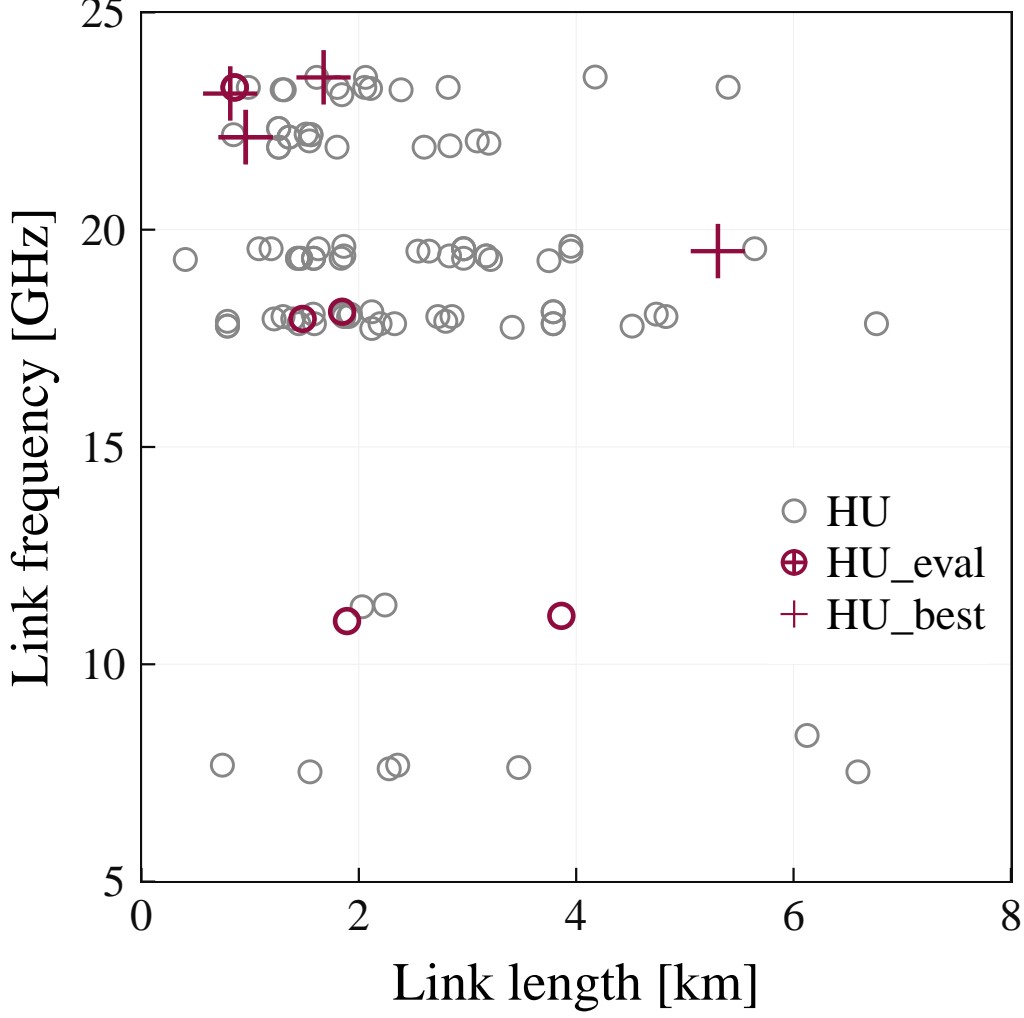

**Figure 2.** Scatter plot of frequency against length for all HU CML (gray circles) shown in Fig. 1. Red markers (either crosses or circles) indicate those 11 CML for which the evaluation was possible. The crosses represent the 5 HU CML which showed clear rainfall signals as compared to nearby gauges, i.e. for which $r^2 \geqslant 0.7$ (Tab. 1, and Sec. 3.1).





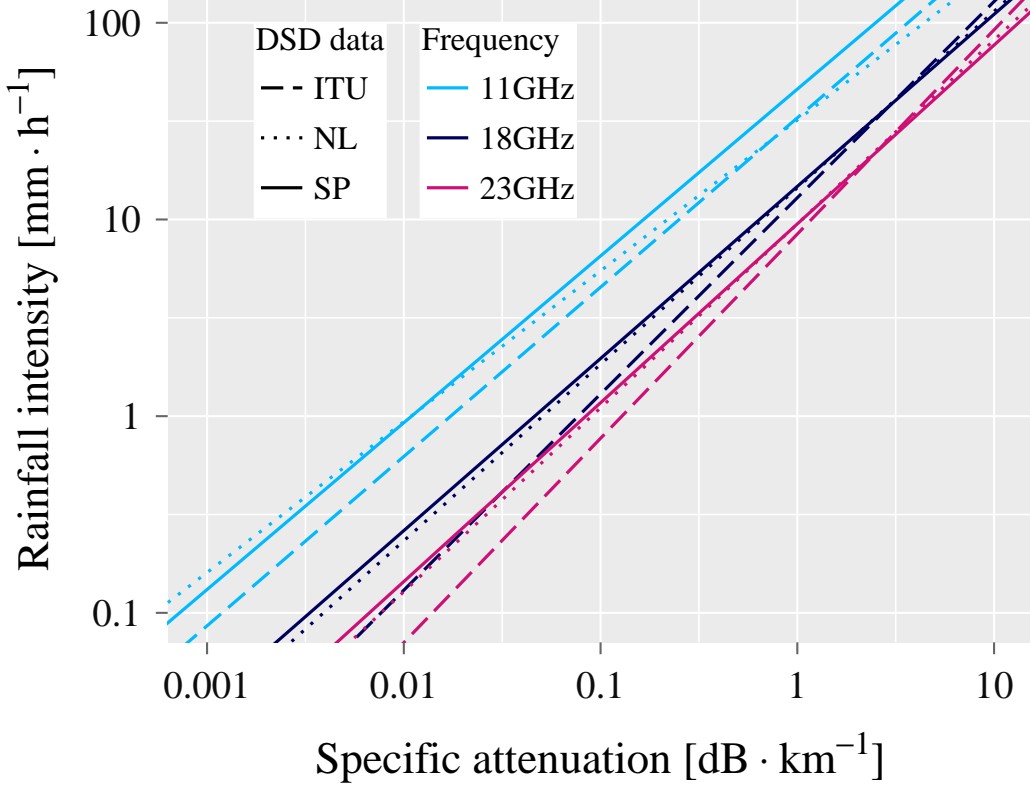

**Figure 3.** Rainfall intensity against specific attenuation for the $a$ and $b$ parameters of 3 DSD models, i.e, local (SP - continuous line), suggested by ITU-R Recommendation P.838-3 (ITU - dashed line), and RAINLINK's default (NL - dotted line). The $R-k$ relations are presented for 3 frequencies: 11 (cyan), 18 (blue), and 23 GHz (pink).





**Figure 4.** Time series of gauge measurements (cyan), RAINLINK estimates (blue), upscaled gauge and CML (dashed and solid green lines, respectively), and minimum and maximum received powers (pink and gold, respectively) for the two longest rainfall events for CML 14 in 2014. The RAINLINK series are computed for the local DSD parameters (Fig. 3, SP $R-k$ relation).





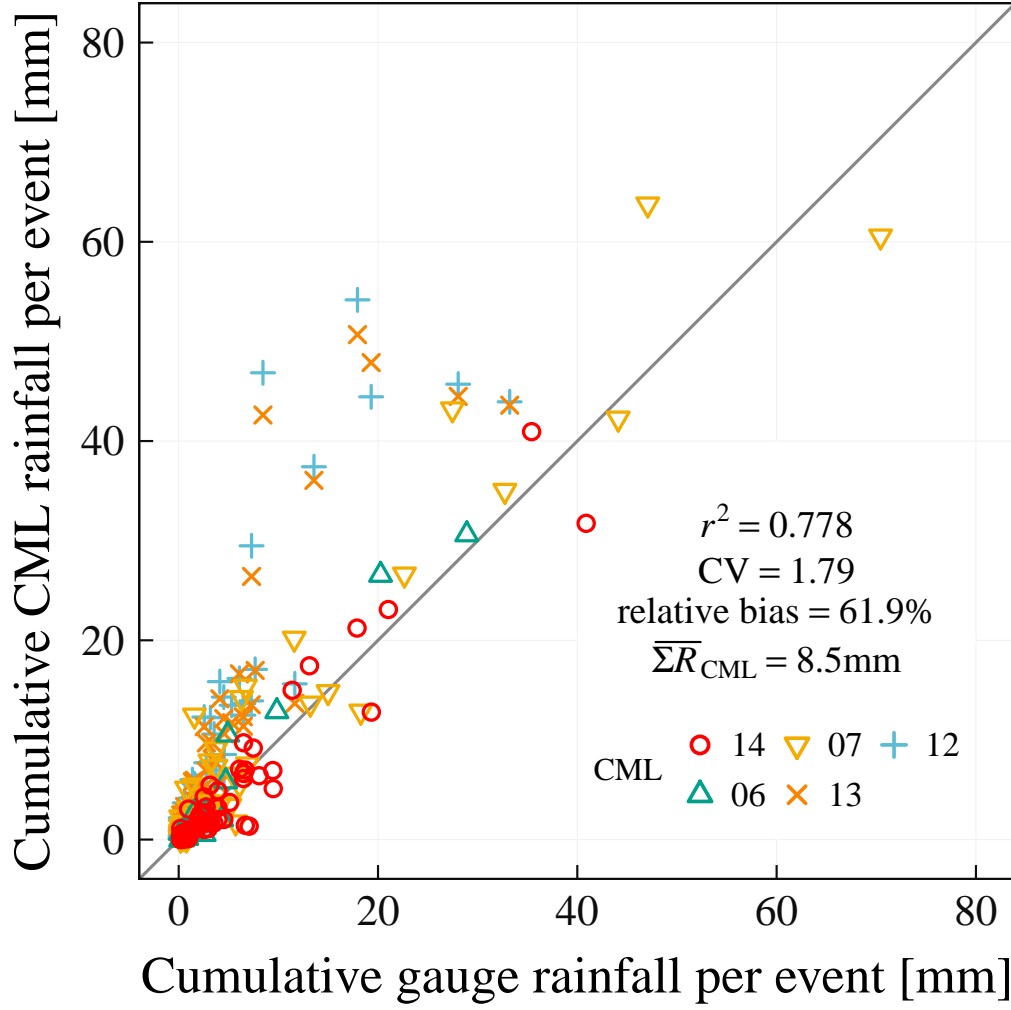

**Figure 5.** Scatter plot of aggregated CML rainfall against aggregated gauge rainfall for all selected rain events (272) in the evaluation of the 5 CML presented in Table 2. RAINLINK estimates computed for the SP $R-k$ relation.





**Table 1.** Relative bias (%), and coefficients of variation (CV) and determination ($r^2$) for $R-k$ relations: local (SP), suggested by ITU-R Recommendation P.838-3 (ITU), and RAINLINK's default (NL). The metrics presented correspond to CML estimates for which $r^2 \geqslant 0.7$ (i.e., 5 out of 11 CML evaluated). n indicates the number of rainfall pairs (30-min intervals) the metrics are computed on. The mean rain gauge depth ($\overline{R}_{\mathrm{gauge}}$) for the n pairs is also presented.

| CML | Length | Freq. | gauge | | $r^2$ | | | CV | | $\overline{R}_{\mathrm{gauge}}$ | | relative bias [%] | | n |
|---|---|---|---|---|---|---|---|---|---|---|---|---|---|---|
| | [km] | [GHz] | | NL | ITU | SP | NL | ITU | SP | [mm] | SP | ITU | NL | |
| 14 | 5.30 | 19.5 | I | 0.738 | 0.746 | 0.735 | 0.72 | 0.70 | 0.72 | 2.07 | - 7.1 | - 15.6 | - 7.3 | 152 |
| 06 | 0.82 | 23.1 | A | 0.924 | 0.923 | 0.923 | 0.77 | 1.03 | 0.68 | 2.41 | 14.9 | 27.3 | 19.9 | 38 |
| 07 | 1.68 | 23.5 | B | 0.851 | 0.852 | 0.849 | 0.86 | 0.99 | 0.83 | 2.70 | 21.9 | 32.2 | 26.6 | 146 |
| 13 | 0.96 | 22.1 | J | 0.809 | 0.803 | 0.809 | 1.97 | 2.44 | 1.79 | 2.08 | 125.1 | 143.0 | 133.8 | 151 |
| 12 | 0.96 | 22.1 | J | 0.801 | 0.793 | 0.801 | 2.29 | 2.83 | 2.08 | 1.97 | 130.9 | 151.7 | 140.3 | 161 |



**Table 2.** Relative bias (%), and coefficients of variation (CV) and determination ($r^2$) for $R-k$ relations: local (SP), suggested by ITU-R Recommendation P.838-3 (ITU), and RAINLINK's default (NL). The metrics presented correspond to significant rainfall events, i.e., most consecutive 30-min steps for which paired rainfall depths are both above $0\,\mathrm{mm}$, and for which their equivalent 15-min minimum received powers were larger than $-90\,\mathrm{dB}$. n indicates the number of significant rainfall events the metrics are computed on.

| CML | Length | Freq. | gauge | $r^2$ | | | CV | | | $\overline{R}_{\mathrm{gauge}}$ | relative bias [%] | | | n |
|---|---|---|---|---|---|---|---|---|---|---|---|---|---|---|
| | [km] | [GHz] | | NL | ITU | SP | NL | ITU | SP | [mm] | SP | ITU | NL | |
| 14 | 5.30 | 19.5 | I | 0.910 | 0.909 | 0.909 | 0.48 | 0.47 | 0.48 | 4.83 | - 7.1 | - 15.6 | - 7.3 | 65 |
| 06 | 0.82 | 23.1 | A | 0.961 | 0.969 | 0.958 | 0.59 | 0.80 | 0.51 | 3.99 | 14.9 | 27.3 | 19.9 | 23 |
| 07 | 1.68 | 23.5 | B | 0.913 | 0.917 | 0.910 | 0.65 | 0.74 | 0.62 | 6.91 | 21.9 | 32.2 | 26.6 | 57 |
| 13 | 0.96 | 22.1 | J | 0.919 | 0.923 | 0.917 | 2.57 | 2.82 | 2.39 | 5.15 | 125.1 | 143.0 | 133.8 | 61 |
| 12 | 0.96 | 22.1 | J | 0.911 | 0.912 | 0.909 | 2.93 | 3.26 | 2.73 | 4.81 | 130.9 | 151.7 | 140.3 | 66 |