# Peer review of "Rainfall retrieval with commercial microwave links in São Paulo, Brazil"

_Atmospheric Measurement Techniques, 2017_

## Referee Comment (RC1) · Anonymous Referee #1 · 11 Oct 2017

**Summary:**

The paper presents an interesting topic. The authors analyze CML data in sub-tropic climate, namely Sao Paulo (Brazil), to derive rainfall information and validate it via a fairly dense network of rain gauges. This seems to be the first time a CML data set from this part of the world is analyzed in this sense, making the manuscript a potentially valuable scientific contribution in AMT. However, in my opinion the analysis is far from complete and misses out a lot of potential. As the authors state, and I acknowledge their honesty, they neglected the majority of the available CML data sets in their analysis, because, either their existing processing code cannot cope with it, or because comparison with nearby rain gauges was not possible or showed low correlation. This is a major shortcoming (see the list of my main concerns below). In general the paper

is well structured and the writing is okay. Given the number of major concerns that I have and owing to the fact that this manuscript is already in open discussion, I recommend a major revision. Completely redoing the analysis with a new direction (focusing more on the CML data quality issue) and resubmitting would maybe be easier if the manuscript would not be openly available already.

**Main concerns:**

- It has already been shown in numerous publications, among them many from the authors of this manuscript, that CML data can be used to derive reliable rainfall information. Hence, the result, that the authors can derive meaningful rainfall information from CML data is not very exciting news. The fact that the rainfall climate is different for the data set presented here, is relevant, however, the impact on the resulting rain rate seems to be negligible in comparison to the other uncertainties (e.g. the considerable differences of the relative bias for the 5 CML-gauge pairs, or the known uncertainties due to wet antenna, quantization, etc.).

- Only being able to derive meaningful results for 5 out of 250 CMLs indicates that either the methods used by the authors are lacking or the technique of using CML data for rainfall estimation in general is less promising than expected.

- The fact that the majority of the CML data, the Ericsson data which only provides the minimum signal levels, cannot be used with the existing codebase of the authors (RAINLINK) should not be an excuse for not analysing it. Rather this calls for adjusting or extending the existing code.

- The final analysis is based only on short or very short CMLs, but the authors do not state if they applied a wet antenna correction method, even though they note themselves that the effect of wet antenna can strongly impact shorter CMLs. This makes all the reasoning about biases arbitrary.

- The authors state that gauge records can also be unreliable, nevertheless they use low correlation with gauge records as indicator to neglect CML data.

**Recommendations:**

- I recommend an extensive major revision, i.e. a real extension of the current analysis (see my points below)

- Given the seemingly very heterogeneous quality of the raw data set (which is fine for an opportunistic sensing technique like the one used here), the scientific focus should in my opinion be to describe how to cope with this data quality issue.

- The constraint to neglect CMLs which are further than 1 km away from a rain gauge should be weakened. One can argue about what a "reasonable" threshold distance for comparing two rainfall measurements is. But, 1 km is really very strict, in particular, since the CMLs integrate over hundreds of meters or several kilometers anyway. The increased distance between CML and gauge will add additional uncertainty for sure, but when I look at the presented results and the relative biases from Table 1, having more data for the analysis seems to be more important than absolute accuracy of rain rates and/or rainfall sums.

- The Ericsson data should be included, i.e. RAINLINK should be extended to be able to process this data, or other code should be written or reused.

**Other major comments and questions:**

Page 4, line 22: What were the actual lengths and frequencies of the "long" CMLs? If the transmit power is high enough or large antennas are used, "uncommon" combination are possible. From Fig 1. some of the very long CMLs look strange indeed, though.

Page 6, line 13: A 50 km radius to look for CMLs with jointly decreasing power levels seems a bit large, in particular since, as the authors write in section 2.1 and 3.1, there is a lot of convective spatially very variable rainfall in the study region. Hence, is this radius of 50km too big? And how sensitive are the RAINLINK processing results on this threshold?

Page 8, line 7: Limiting the analysis to CML-gauge pairs were both show a rainfall depth above 0 mm, neglects the validation of the challenging step of detecting rain events in the CML time series, which, to my understanding, is the first step in RAINLINK. Wrong detections, i.e. missed rain events or artificially generated rain, may considerably add bias to the accumulations. Hence, this effect should be included in the validation or added in a separate validation.

Page 8, line 31ff: Given that this is the result for 1 out of 250 CMLs, I would recommend not to draw that optimistic conclusions based on the current state of the analysis.

Fig 1: As it is mentioned in the text, the very long CMLs indeed look strange since they do not even end on one of the summit of the mountains in the north and north-east. Wouldn't it be possible to check via GoogleMaps satellite images if there is a relay or cell phone tower there? It would be nice to have a more solid basis for neglecting these CMLs. At least give more details in the text. Maybe it would also be good to show two or three maps, one with all CMLs, one with "reasonable" CMLs and in addition only the CMLs used for analysis (which hopefully will be much more in the next revision of the manuscript. . .).

**Technical and minor comments** (this is a uncomplete list, since I assume that the manuscript will considerably change with the next iteration):

Fig 2: I only see 4 crosses not 5 as indicated in the caption. Also the red circles and red crosses seem not to add up to 11. Maybe overplotting is an issue here. If yes, this should be mentioned. Furthermore, no CMLs longer than 8 km are shown, even though the caption states that all HU CMLs are plotted, for which, according to Fig 1.,

some are definitely longer than 8 km.

Fig 5: The two yellowish colors are hard to distinguish. Anyway, if colors are different, markers could maybe be the same to make the graph easier to read. Or even better, have separate scatter plots for the CMLs, or at least for selected ones, if the number of CMLs increases with an extended analysis.

Table 1 and Table 2: The relative biases are exactly the same in both tables. As far as I understood, Table 2 is based only on a subset of the rain events from Table 1. Hence, I assume there is something wrong with either Table 1 or Table 2.

Table 1 and Table 2: Is CML 12 and 13 along the same path, but just the two directions?

---

## Referee Comment (RC2) · Anonymous Referee #2 · 15 Oct 2017

The paper proposes to analyze an important topic : possible use of CMLs data for quantitative rainfall estimation in one of the largest city under tropical climate, i.e Sao Paulo. As reminded by the authors the area is prone to intense rainfall, leading to flash floods and other natural hazards such as land slides.

The authors and various other groups have already demonstrated the potential of the CMLs based method under a range of climate and weather situations ( from widespread systems in the Netherland to intense convection in Africa, through mediter-ranean areas and even mountainous regions). This new data set in Brazil is an opportunity to test the CMLs method in a more challenging context then in previous studies : the quality of the CMLs data set is not homogenous, the validation network is sparse. The authors seems to have partially avoided this challenge by focusing only on a very

limited subsample of the data set (where and when it works....); unfortunately this also limits the scientific impact of the study and its interest as a demonstrator of CMLs potential for hydro-meteorological monitoring over Sao Paulo....

Given the existing literature on the CMLs topic and the extensive data set available here, the present study should be taken a step further and provide a more robust and extensive analysis of the available data set, including issues related to data quality, sparse GV and data format variation among CML providers..

A major limitation of the paper in its present form is that conclusions are drawn from a very limited subset of the available data set : only a few links (5 out of a possible total of above 200 ) are exploited and for theses links the analysis is restricted to time steps where both the link and the nearby gauge detect rainfall. Doing so the authors miss a major issue : capability of the method to detect rain and not generate false alarms, and so over the whole network.

This a major forthcoming of an otherwise very well written paper, which also provides a good review of the state of the art in CMLs based rainfall estimation. I can only encourage the authors to take the necessary time to submit a improved version of their work and take the analysis a step further.

Detailed major/minor recommendations :

-One important feature of sub-tropical rainfall is the occurrence of intense (and possibly extreme) rainfall rates associated with convective cells. This is very important for some of the applications the authors put forward in their introduction . No information is provided on the actual rain rate distribution (at the 30 minutes time step for instance) observed over the study period in Sao Paulo by the gauges and how well ( or not) the CML method retrieves it. The global statistics provided in Table 2 and 3 do not inform us on the performance of the Rainlink/CML data according to rain rate classes . This is an important question, for hydrological applications for instance.

-Selection of the time steps and 'events'. The authors should provide statistics covering the whole analysis period and not solely on a selected number of 30' times steps. Time step where one OR the other sensor detected rain should be included and a contingency table provided. The definition of 'events' , as presented in Fig 5 is not clear. Does it include some non rainy time steps or is it based on the same selection as the 30 ' (both CMLS and gauge > 0)? Daily statistics would be useful and would allow comparisons with other studies . . ..

-The authors mention wet antenna as a possible source of bias : this should be explored further - The order of magnitude of wet antenna attenuation is known, is it compatible with the observed bias ?

-CMLS data selection : the authors should extend the analysis to other CMLs links even if they keep the present 5 links to illustrate the best case– This is important to asses the actual potential of the method in a context representative of reality. Given that the analysis is carried out at the 30' and 'event' time step, 1 km maximum distance from the gauge seems very severe.

The conclusions should be revised once a truly extensive assessment has been done on this data set.

I am looking forward to see a revised version that will investigate further this rich data set acquired in Brazil !

―――――――――――――――――――――

---

## Referee Comment (RC3) · Anonymous Referee #3 · 17 Oct 2017

**General comments:**

The manuscript Rainfall retrieval with commercial microwave links in Sao Paulo Brazil aims to evaluate potential of commercial microwave links (CMLs) as rainfall sensors in the subtropical climate. The authors collect data from several microwave links, process them using RAINLINK R package and compare them to rain gauges operated by CAMADEN. Moreover, disdrometer observations are used to estimate parameters of attenuation-rainfall power-law model and these parameters are compared to those from ITU recommendations and from Dutch case studies.

Although the topic of CML rainfall retrieval in subtropical climate is relevant and the presented dataset is valuable the study has several major drawbacks: i) the authors select for evaluation only well performing CMLs. This is a reasonable approach if the

selection procedure is independent of a reference rainfall data. However, this is not the case, as one of the selection criteria is correlation of CMLs to the reference rain gauges ii) the results are presented and discussed very briefly without sufficient attempt to investigate the causes of good/bad performance of particular CMLs. The influence of drop size distribution to the attenuation-rainfall model is analyzed in more detail, however, this effect can explain only small fraction of total errors. Especially spatial representativeness of reference rain gauge data should be more properly analyzed to avoid interpreting discrepancy between path-integrated CML rainfall observation and point RG rainfall observation as an inaccuracy of a CML iii) The conclusions are not sufficiently supported by the data. The authors claim that CMLs are very promising source of rainfall data only based one very good and two relatively well performing CMLs. Also the suitability of RIANLINK package for CML processing in subtropical regions is not proofed. The data rather indicate that constant WAA correction used in the RAINLINK package is inappropriate for CMLs.

Given the above mentioned shortcomings the reviewer does not recommend the manuscript for a publication, however, encourage the authors to improve the data analysis, rewrite especially the results, discussion and conclusion sections and resubmit the manuscript. Some suggestions for revisions are given in the specific comments bellow.

**Specific comments:**

The reviewer suggests changing the structure of the manuscript: i) moving the descriptions of the evaluation procedure (event definition) from the Results and Discussion section to the Data and Methods section, ii) considering moving the results of DSD analysis from the Rainfall retrieval algorithm section.

**P4L26:** Is the threshold value r2 ≥ 0.7 chosen arbitrary? Why not 0.5 or 0.9? In any case, the selection of CMLs for evaluation based on reference data does not enable to evaluate potential of CMLs without having reference rainfall. This is one of the major drawbacks of the whole analysis. Moreover, it might be valuable keeping the bad performing CMLs in the analysis and identify the causes of the bad performance.

**P5L5:** Given the CML paths lengths from several hundreds of meters up to several km the criterion of 1 km distance from link path seems to be too strict and not always reasonable. E.g. for CML 14 it might be more representative to use average of two RGs even though the second RG is several km far away. In any case, the reviewer suggests presenting at least some basic analysis of RG correlation and set the criterion based on this analysis. Such analysis would also support the results and enable to distinguish between discrepancy of path and point measured rainfall and between errors due to inaccuracy of CMLs.

**P6L21:** The section describes rather in detail generally well known performance metrics, however does not provide complete information about evaluation procedure. E.g. it should be explained here how the event based evaluation is performed (metrics are calculated for each event and then averaged as presented in Tab 2?).

**P7L15:** why -90 dB and not some other value?

**P7L17:** Both overall evaluation and event based evaluation is presented here. This is very good idea, as one could learn e.g. during which types of events CMLs perform well. However, at the end the event based results are presented in overall statistics (Tab. 2) except results presented in the Fig. 5. It might be very interesting to see how stable the CML performance is (e.g. in terms of variance of the metrics). This could be presented as boxplots or scatter plots of metrics, similarly as on Fig. 5. This would also enable more proper discussion of the results with potentially answering to questions like these: Do CMLs perform better during strong rainfalls than light rainfalls? Do they better reproduce rainfall temporal dynamics (r2) during light or heavy rainfalls?

**P8L6-L11:** The event definition might be rather in the method section

**P8L14-18:** It seems that shorter CMLs are substantially more biased than longer

CMLs. This indicates that the bias arises from wet antenna attenuation. Thus, RAIN-LINK's representation of baseline (constant) seems not working very well.

**P8LL35 – P9L2:** The performance was clearly very good only for one CML whereas the other experience relatively high bias. This is not really proving the good performance of RAINLINK in subtropical regions.

**P9L18-20 and P10L4-6:** Only three CMLs out of 17 resp. 11 were identified (based on reference rainfall) as well performing. The suitability of RAINLINK for processing such data should be, therefore, discussed more critically. Similarly, the authors claim that the potential of CMLs would be great if the data and metadata are properly stored. This is unfortunately not happening in the reality as demonstrated by the presented results. Thus, use of CMLs for subtropical regions is still rather big challenge. The dataset presented in this paper might, however, contribute to coping with this challenge. Thus, the reviewer highly encourages the authors to invest more work into its analysis and resubmit the improved manuscript.

**Fig.1:** CMLs selected for the analysis are really tiny in the figure. Maybe cropping and resizing the figure would help (long CMLs aiming to the north-west could be cropped as they are not used for the analysis).

**Tab. 2:** It seems to be there is no distinctive difference in the effect of DSD when evaluated over the whole dataset (tab. 1) and event based. It might be, therefore, reasonable to present here only results for fitted DSD (i.e. best performing a, b parameters) and instead one value (Mean of a metric?) present e.g. mean and standard deviation of a metric.

---

## Author Comment (AC1) · 30 Nov 2017

**Summary:**

The paper presents an interesting topic. The authors analyze CML data in sub-tropic climate, namely Sao Paulo (Brazil), to derive rainfall information and validate it via a fairly dense network of rain gauges. This seems to be the first time a CML data set from this part of the world is analyzed in this sense, making the manuscript a potentially valuable scientific contribution in AMT. However, in my opinion the analysis is far from complete and misses out a lot of potential. As the authors state, and I acknowledge their honesty, they neglected the majority of the available CML data sets in their analysis, because, either their existing processing code cannot cope with it, or because comparison with nearby rain gauges was not possible or showed low correlation. This is a major shortcoming (see the list of my main concerns below). In general the paper is well structured and the writing is okay. Given the number of major concerns that I have and owing to the fact that this manuscript is already in open discussion, I recommend a major revision. Completely redoing the analysis with a new direction (focusing more on the CML data quality issue) and resubmitting would maybe be easier if the manuscript would not be openly available already.

R/. We thank the reviewer for the constructive assessment of our manuscript.

**Main concerns:**

• It has already been shown in numerous publications, among them many from the authors of this manuscript, that CML data can be used to derive reliable rainfall information. Hence, the result, that the authors can derive meaningful rainfall information from CML data is not very exciting news. The fact that the rainfall climate is different for the data set presented here, is relevant, however, the impact on the resulting rain rate seems to be negligible in comparison to the other uncertainties (e.g. the considerable differences of the relative bias for the 5 CML-gauge pairs, or the known uncertainties due to wet antenna, quantization, etc.).

**R/.** The fact that for this dataset the impact of a different rainfall climate indeed seems limited can also be interpreted as an import finding. It gives an indication that the rainfall retrieval from CML data is relatively insensitive to the rainfall climate, which confirms its general applicability. In addition, obtaining CML data is generally regarded as difficult. In this study CML data from, Brazil have been obtained, which shows that cellular communication companies are willing to provide data for yet another region.

• Only being able to derive meaningful results for 5 out of 250 CMLs indicates that either the methods used by the authors are lacking or the technique of using CML data for rainfall estimation in general is less promising than expected.

**R/.** We acknowledge that our rainfall retrieval algorithm, developed for a temperate climate, may not be optimal for another climate, although we expect that the main principles, e.g. those behind the wet-dry classification, still hold. Rainfall estimation using CML data has already been successfully demonstrated for a variety of climates (Mediterranean, temperate, mountainous, and a tropical climate), although not for a subtropical one and not always on large datasets. For some seasons the rainfall characteristics from these climates will show similarities with a subtropical climate. Hence, we do expect that CML data are also promising for rainfall estimation in a subtropical climate. Please note that 35 out of 101 HU CMLs

could not be used because of lacking received signal levels due to issues in the log-file of the attenuation measurements. Moreover, 149 CMLs (ER) were not used because only minimum received powers were provided (which we will analyze in a revised version of this paper; see our response to the next comment). As far as we know, such a sampling strategy has not been encountered before by the microwave link rainfall estimation community, and RAINLINK has not been optimized for this. So actually we end up with 5 out of 66, instead of 250, CMLs providing reasonable estimates. Only 17 CMLs are within 1 km of a rain gauge, of which 6 CMLs were discarded due to dubious metadata (a strange microwave frequency and path length combination), and 6 other ones showing low correlation with rain gauge data perhaps because of erroneous metadata. So the number of CMLs revealing a clear rain signal could be (much) higher than 5 if those 149 CMLs only recording minimum received signal levels and those 49 more than 1 km away from a rain gauge would have been analyzed. Data availability and erroneous metadata certainly play a role. In a revised version of the manuscript we will greatly extend the analyses to include as many CML data as possible. Firstly, we will analyze the 149 Ericsson links as well (see our response to the next reviewer comment). And secondly, we will show rainfall accumulations for all links (also those that do not have a gauge within 1 km). Note that we will use a simple quality check on the gauges before we decide to use them in further analyses, and that we will remove all links that have unrealistic lengthfrequency combinations from the analyses.

• The fact that the majority of the CML data, the Ericsson data which only provides the minimum signal levels, cannot be used with the existing codebase of the authors (RAINLINK) should not be an excuse for not analysing it. Rather this calls for adjusting or extending the existing code.

**R/.** We agree with the reviewer that it would be valuable to also obtain rainfall estimates from CMLs providing only minimum received powers. It appears that the RAINLINK package itself does not need any modifications to do so. We only need to set Pmax to Pmin in a preprocessing step, and then use a conversion factor from maximum rainfall intensity to average rainfall intensity as a postprocessing step. Hence, we will show the analyses of the Ericsson dataset from its processing through RAINLINK in a revised version of our manuscript, as the reviewer encourages us now. The number of useful CMLs is expected to be much lower than 149. For instance, out of those 149 CMLs, many will be discarded because of the selection stated on page 4 of our manuscript "Our experience tells us that CML with both lengths above 20 km and frequencies above 15 GHz are not common in CML networks (they are highly unlikely from a network design perspective: long links experience more attenuation in rain, and should hence operate at low frequencies to limit this attenuation).". A preliminary and equivalent figure for the Ericsson dataset. Nevertheless, it is certainly worthwhile to also employ the Ericsson links.

\* The orange circle indicates the region where reliable measurements can be drawn from. The region where L>20 km and F>15 GHz is clearly where the unrealistic length-frequency combinations are. We also choose to not include those links with frequencies below 15 GHz because these frequencies are less optimal for CML rainfall retrieval.

• The final analysis is based only on short or very short CMLs, but the authors do not state if they applied a wet antenna correction method, even though they note themselves that the effect of wet antenna can strongly impact shorter CMLs. This makes all the reasoning about biases arbitrary.

*R/.* We do apply a fixed wet antenna attenuation correction as described in Overeem et al. (2016a), using the default value of 2.3 dB. We will add this to the main text.

• The authors state that gauge records can also be unreliable, nevertheless they use low correlation with gauge records as indicator to neglect CML data.

**R/.** We agree with the suggested concept of "false negative" that the reviewer implies. Nevertheless, and given that rain gauges are the only available source we could refer our CMLs rain retrievals to, we considered that "it is likely that the high values of r2 indicate that both types of observations contain a true rain signal." (page 8). With lower correlations, it's actually not possible to know whether the inaccurate rainfall estimate comes from the CML or gauge measurements. In the revised version of our paper, we will use a basic quality check on the gauge data before using the gauges in the analyses. This quality check is based on comparison of (long-term) accumulations with those from its nearest neighbors.

**Recommendations:**

• I recommend an extensive major revision, i.e. a real extension of the current analysis (see my points below)

**R/.** We will present analyses on a greatly extended datasets in the revised version of the paper.

• Given the seemingly very heterogeneous quality of the raw data set (which is fine for an opportunistic sensing technique like the one used here), the scientific focus should in my opinion be to describe how to cope with this data quality issue.

**R/.** To provide suggestions on how to cope with the data quality issue is rather difficult given the errors in the metadata and the lack of a study confirming the quality of our reference rain gauge data. However, we do give recommendations on using link length and frequency information for link metadata quality control. And the RAINLINK package also includes several quality control steps. The analyses presented here are also an indication to what degree these quality control procedures work.

• The constraint to neglect CMLs which are further than 1 km away from a rain gauge should be weakened. One can argue about what a "reasonable" threshold distance for comparing two rainfall measurements is. But, 1 km is really very strict, in particular, since the CMLs integrate over hundreds of meters or several kilometers anyway. The increased distance between CML and gauge will add additional uncertainty for sure, but when I look at the presented results and the relative biases from Table 1, having more data for the analysis seems to be more important than absolute accuracy of rain rates and/or rainfall sums.

**R/.** We agree with the reviewer that the 1-km constraint removes a large part of the links. On the other hand, we would like to limit the influence of sampling errors on the analyses. In order to meet both of these requirements we present an additional global analysis where we show time series of rainfall retrievals from all links, compared to the averaged gauge accumulations. For the computation of statistics we will keep the 1-km constraint. In this way, we will show the potential of this method on a large dataset, as well as limiting the effect of sampling differences on the more quantitative analyses.

• The Ericsson data should be included, i.e. RAINLINK should be extended to be able to process this data, or other code should be written or reused.

**R/.** We'll include the analysis of the Ericsson dataset (see replies above).

**Other major comments and questions:**

Page 4, line 22: What were the actual lengths and frequencies of the "long" CMLs? If the transmit power is high enough or large antennas are used, "uncommon" combination are possible. From Fig 1. some of the very long CMLs look strange indeed, though.

*R/.* Please see Figure 1, where frequencies against link-lengths are shown for all possible CMLs in our dataset. Not shown in this figure are several 100-km CMLs with frequencies close to 0. Personal communication with network design engineers from a telecommunication company in the Netherlands, confirms that the discarded microwave frequency - path length combinations should be erroneous. Having a larger transmit power to compensate for longer path lengths is generally not used because of the greatly increased probability of interference with other systems in the same band.

Page 6, line 13: A 50 km radius to look for CMLs with jointly decreasing power levels seems a bit large, in particular since, as the authors write in section 2.1 and 3.1, there is a lot of convective spatially very variable rainfall in the study region. Hence, is this radius of 50km too big? And how sensitive are the RAINLINK processing results on this threshold?

**R/.** Figures 2 and 3 (below) show the "in-sensitivity" of the analyses to either the use a radius-threshold ("rOF" approach) or the no use of it at all ("rPP") approach. Figure 3 is equal to Figure 2 but all 16 CML in the analysis are plotted in the box-plots. In the revised version of the manuscript we will consider using a

shorter radius (15 or 20 km). Given the fact that we intend to use more links in the analyses, this should be feasible.

rPP

400 -350 -300 -

250 -

200 -

150 -100 -50 --50 --100 -

rÖF

r2

| dataSet | linkID   |        |   |        |
|---------|----------|--------|---|--------|
| rOF     | 0        | 060P01 | ٠ | 126C01 |
|         |          | 060P06 | ⊕ | 134N01 |
|         | +        | 074N01 | 歞 | 142A02 |
|         | ×        | 089A02 | • | 142N01 |
|         | 0        | 089P05 | 8 | 145V01 |
|         | $\nabla$ | 105F01 |   | 148N01 |
|         |          | 110A04 |   | 153001 |
|         | *        | 124A02 | • | 158001 |
|         |          |        |   |        |

Page 8, line 7: Limiting the analysis to CML-gauge pairs were both show a rainfall depth above 0 mm, neglects the validation of the challenging step of detecting rain events in the CML time series, which, to my understanding, is the first step in RAINLINK. Wrong detections, i.e. missed rain events or artificially generated rain, may considerably add bias to the accumulations. Hence, this effect should be included in the validation or added in a separate validation.

**R/.** We agree with the reviewer that this is indeed an important aspect of CML rainfall retrieval. We will therefore include analyses of this aspect in the revised version of the paper.

Page 8, line 31ff: Given that this is the result for 1 out of 250 CMLs, I would recommend not to draw that optimistic conclusions based on the current state of the analysis. *R/.* We will rephrase our conclusions if needed based on the analyses of the greatly extended CML dataset.

Fig 1: As it is mentioned in the text, the very long CMLs indeed look strange since they do not even end on one of the summit of the mountains in the north and north-east. Wouldn't it be possible to check via GoogleMaps satellite images if there is a relay or cell phone tower there? It would be nice to have a more solid basis for neglecting these CMLs. At least give more details in the text. Maybe it would also be good to show two or three maps, one with all CMLs, one with "reasonable" CMLs and in addition only the CMLs used for analysis (which hopefully will be much more in the next revision of the manuscript: : :).

**R/.** Checking the locations of the antennas of links on e.g. Google Maps could indeed be a valuable addition. However, the effort of manually checking antenna locations is not feasible for the large dataset we're dealing with here. It is also important to realize that antennas that were previously used for other links could have been re-used without having changed the location metadata in the database (a likely error; personal communication with representatives from a cellular communication company in the Netherlands). This means that there are likely still antennas at that location, but the specific antenna will have been moved. Hence, checking for the presence of antennas on Google Maps is likely not to yield the necessary information.

**Technical and minor comments** (this is a uncomplete list, since I assume that the manuscript will considerably change with the next iteration):

Fig 2: I only see 4 crosses not 5 as indicated in the caption. Also the red circles and red crosses seem not to add up to 11. Maybe overplotting is an issue here. If yes, this should be mentioned. Furthermore, no CMLs longer than 8 km are shown, even though the caption states that all HU CMLs are plotted, for which, according to Fig 1., some are definitely longer than 8 km.

*R/.* This is indeed due to overplotting, which will be mentioned in the caption.

Huawei CMLs with no useful data (or not close to gauges) could be larger than 8km (see Figure 1 of this rebuttal). But Figure 2 only shows the 11 selected CMLs, which are all shorter than 8 km. In the revised version of the manuscript, we will extend the axes of this figure to include all links that were used in the analyses.

Fig 5: The two yellowish colors are hard to distinguish. Anyway, if colors are different, markers could maybe be the same to make the graph easier to read. Or even better, have separate scatter plots for the CMLs, or at least for selected ones, if the number of CMLs increases with an extended analysis. *R/.* The reason for using different markers for the different links in this graph is indeed that some of the colors could be difficult to distinguish. We feel that the use of different markers is sufficient to make the graph readable. We will consider making separate plots if this is required due to the extended dataset.

Table 1 and Table 2: The relative biases are exactly the same in both tables. As far as I understood, Table 2 is based only on a subset of the rain events from Table 1. Hence, I assume there is something wrong with either Table 1 or Table 2.

**R/.**The relative biases presented in Tables 1 and 2 should be exactly the same. Table 1 presents the statistics for all 30-minute accumulations, whereas Table 2 presents the same statistics for event accumulations (where an event is a contiguous period in time with nonzero rainfall). Because the bias is the difference (between link estimates and gauge measurements) in total accumulated rain (normalized by the gauge accumulation), this bias should be exactly the same in both tables (because the total accumulations are also exactly the same). The difference between the two tables is in the other statistics.

Table 1 and Table 2: Is CML 12 and 13 along the same path, but just the two directions? **R/.** Exactly. We will mention this in the revised version of the paper.

**Anonymous Referee #2**

Received and published: 15 October 2017

The paper proposes to analyze an important topic : possible use of CMLs data for quantitative rainfall estimation in one of the largest city under tropical climate, i.e Sao Paulo. As reminded by the authors the area is prone to intense rainfall, leading to flash floods and other natural hazards such as land slides. The authors and various other groups have already demonstrated the potential of the CMLs based method under a range of climate and weather situations (from widespread systems in the Netherland to intense convection in Africa, through Mediterranean areas and even mountainous regions). This new data set in Brazil is an opportunity to test the CMLs method in a more challenging context then in previous studies: the quality of the CMLs data set is not homogenous, the validation network is sparse. The authors seems to have partially avoided this challenge by focusing only on a very limited subsample of the data set (where and when it works: : :.); unfortunately this also limits the scientific impact of the study and its interest as a demonstrator of CMLs potential for hydro-meteorological monitoring over Sao Paulo: : :.

Given the existing literature on the CMLs topic and the extensive data set available here, the present study should be taken a step further and provide a more robust and extensive analysis of the available data set, including issues related to data quality, sparse GV and data format variation among CML providers..

**R/.** We thank the reviewer for the constructive review. We will extend the analyses by including the 149 Ericsson links, from which only minimum received powers are available. And we will present the rainfall retrievals of the links further than 1 km from a gauge in a separate analysis. In many cases it will be difficult to investigate the data quality in more detail, because of erroneous metadata or the lack of information regarding the quality of the rain gauge network used as a reference. We will, however, use a basic quality check on the gauge data to remove malfunctioning gauges. Please see our replies to reviewer #1 for more details.

A major limitation of the paper in its present form is that conclusions are drawn from a very limited subset of the available data set : only a few links (5 out of a possible total of above 200) are exploited and for theses links the analysis is restricted to time steps where both the link and the nearby gauge

detect rainfall. Doing so the authors miss a major issue : capability of the method to detect rain and not generate false alarms, and so over the whole network.

**R/.** In the revised version of the paper we will present analyses of a greatly extended dataset, and we will adapt the conclusions if necessary. Furthermore, we will add information about the effectiveness of estimating zero rain (see our replies to reviewer #1 for more details).

This a major forthcoming of an otherwise very well written paper, which also provides a good review of the state of the art in CMLs based rainfall estimation. I can only encourage the authors to take the necessary time to submit a improved version of their work and take the analysis a step further.

Detailed major/minor recommendations :

-One important feature of sub-tropical rainfall is the occurrence of intense (and possibly extreme) rainfall rates associated with convective cells. This is very important for some of the applications the authors put forward in their introduction . No information is provided on the actual rain rate distribution (at the 30 minutes time step for instance) observed over the study period in Sao Paulo by the gauges and how well ( or not) the CML method retrieves it. The global statistics provided in Table 2 and 3 do not inform us on the performance of the Rainlink/CML data according to rain rate classes . This is an important question, for hydrological applications for instance.

**R/.** We agree with the reviewer that this is indeed a relevant question. However, we feel that this is outside of the scope of the present paper and a topic for future research. We will state this as a recommendation for future work in the revised version of the paper.

-Selection of the time steps and 'events'. The authors should provide statistics covering the whole analysis period and not solely on a selected number of 30' times steps. Time step where one OR the other sensor detected rain should be included and a contingency table provided. The definition of 'events', as presented in Fig 5 is not clear. Does it include some non rainy time steps or is it based on the same selection as the 30 ' (both CMLS and gauge > 0)? Daily statistics would be useful and would allow comparisons with other studies : : :.

**R/.** We decided to provide conditional statistics (i.e. R > 0 mm/h) on CML rainfall retrieval performance. In the revised version of the paper we will also include information about the effectiveness of estimating zero rain (see our responses to the comments by reviewer #1 for more details).

-The authors mention wet antenna as a possible source of bias : this should be explored further - The order of magnitude of wet antenna attenuation is known, is it compatible with the observed bias ? **R/.** We already apply a fixed wet antenna attenuation correction of 2.3 dB. This is just an average value. For a chosen rainfall event, the wet antenna attenuation may differ a lot since one, two or no antennas can become wet. Moreover, the amount of attenuation can also depend on rainfall intensity, and biases can also be caused by other phenomena. Hence, it is rather difficult to assess whether remaining biases are caused by wet antenna attenuation. At least part of the wet antenna attenuation has been corrected for.

-CMLS data selection : the authors should extend the analysis to other CMLs links even if they keep the present 5 links to illustrate the best case— This is important to asses the actual potential of the method in a context representative of reality. Given that the analysis is carried out at the 30' and 'event' time step, 1 km maximum distance from the gauge seems very severe.

**R/.** In the revised version of this paper we will present analyses with a greatly extended dataset of CMLs. More details of the additional analyses can be found in our replies to the comments of reviewer #1.

The conclusions should be revised once a truly extensive assessment has been done on this data set. I am looking forward to see a revised version that will investigate further this rich data set acquired in Brazil !

**R/.** We will modify our conclusions based on the analyses of the much larger dataset if necessary.

**Anonymous Referee #3**

Received and published: 17 October 2017

**General comments:**

The manuscript Rainfall retrieval with commercial microwave links in Sao Paulo Brazil aims to evaluate potential of commercial microwave links (CMLs) as rainfall sensors in the subtropical climate. The authors collect data from several microwave links, process them using RAINLINK R package and compare them to rain gauges operated by CAMADEN. Moreover, disdrometer observations are used to estimate parameters of attenuation-rainfall power-law model and these parameters are compared to those from ITU recommendations and from Dutch case studies.

Although the topic of CML rainfall retrieval in subtropical climate is relevant and the presented dataset is valuable the study has several major drawbacks: i) the authors select for evaluation only well performing CMLs. This is a reasonable approach if the selection procedure is independent of a reference rainfall data. However, this is not the case, as one of the selection criteria is correlation of CMLs to the reference rain gauges ii) the results are presented and discussed very briefly without sufficient attempt to investigate the causes of good/bad performance of particular CMLs. The influence of drop size distribution to the attenuation-rainfall model is analyzed in more detail, however, this effect can explain only small fraction of total errors. Especially spatial representativeness of reference rain gauge data should be more properly analyzed to avoid interpreting discrepancy between path-integrated CML rainfall observation and point RG rainfall observation as an inaccuracy of a CML iii) The conclusions are not sufficiently supported by the data. The authors claim that CMLs are very promising source of rainfall data only based one very good and two relatively well performing CMLs. Also the suitability of RIANLINK package for CML processing in subtropical regions is not proofed. The data rather indicate that constant WAA correction used in the RAINLINK package is inappropriate for CMLs.

Given the above mentioned shortcomings the reviewer does not recommend the manuscript for a publication, however, encourage the authors to improve the data analysis, rewrite especially the results, discussion and conclusion sections and resubmit the manuscript. Some suggestions for revisions are given in the specific comments bellow.

**Specific comments:**

The reviewer suggests changing the structure of the manuscript: i) moving the descriptions of the evaluation procedure (event definition) from the Results and Discussion section to the Data and Methods section, ii) considering moving the results of DSD analysis from the Rainfall retrieval algorithm section.

**R/.** We feel that the description of the evaluation procedure is an important part of the Results and Discussions section, and that moving this part would not increase the readability of the paper. The same holds for the DSD analyses and the Rainfall Retrieval Algorithm section. We will therefore keep the structure of the paper as it was.

**P4L26:** Is the threshold value r2 \_ 0.7 chosen arbitrary? Why not 0.5 or 0.9? In any case, the selection of CMLs for evaluation based on refer

---

## Author Response (AR1)

**Summary:**
The paper presents an interesting topic. The authors analyze CML data in sub-tropic climate, namely Sao Paulo (Brazil), to derive rainfall information and validate it via a fairly dense network of rain gauges. This seems to be the first time a CML data set from this part of the world is analyzed in this sense, making the manuscript a potentially valuable scientific contribution in AMT. However, in my opinion the analysis is far from complete and misses out a lot of potential. As the authors state, and I acknowledge their honesty, they neglected the majority of the available CML data sets in their analysis, because, either their existing processing code cannot cope with it, or because comparison with nearby rain gauges was not possible or showed low correlation. This is a major shortcoming (see the list of my main concerns below). In general the paper is well structured and the writing is okay. Given the number of major concerns that I have and owing to the fact that this manuscript is already in open discussion, I recommend a major revision. Completely redoing the analysis with a new direction (focusing more on the CML data quality issue) and resubmitting would maybe be easier if the manuscript would not be openly available already.
*R/. We thank the reviewer for the constructive assessment of our manuscript.*

**Main concerns:**
• It has already been shown in numerous publications, among them many from the authors of this manuscript, that CML data can be used to derive reliable rainfall information. Hence, the result, that the authors can derive meaningful rainfall information from CML data is not very exciting news. The fact that the rainfall climate is different for the data set presented here, is relevant, however, the impact on the resulting rain rate seems to be negligible in comparison to the other uncertainties (e.g. the considerable differences of the relative bias for the 5 CML-gauge pairs, or the known uncertainties due to wet antenna, quantization, etc.).
*R/. Through a more comprehensive analysis we found a suitable application of RAINLINK for a subtropical climate like the one of São Paulo, despite of RAINLINK being calibrated for a typical Dutch climatology. The suitable applicability of RAINLINK for such (subtropical) climatologies comes from the fact that we were able to identify its "alpha" parameter as 0.38, which is rather similar to the default one of 0.33 (which was obtained from Dutch rainfall data). Thus, as the reviewer suggests, this has indeed a negligible impact on the RAINLINK output. Nevertheless, following the same comprehensive analysis, we were able to identify the three best performing CMLs (Commercial Microwave Links), one of which is an ER (Ericsson) CML, which is three times more than we were able to identify for the first version of the manuscript. The particularity of these three best CMLs is that they are shorter than 1.7 km, where representativeness errors play a smaller role. We also found overestimations in CML estimates (Figs. 4 and 6 of the updated version of the manuscript). Such overestimations may be related to the fact that rain-induced attenuation along the link path may be relatively small compared to the attenuation caused by wet antennas, i.e., the wet antennas could contribute to some of the overestimations.*
*Hence, and in the updated version, we now reflect on this stating that "The results of Fig. 5 are obtained for short links (< 1.7 km), where representativeness errors will play a smaller role. Overestimations by CMLs may be related to the fact that rain-induced attenuation along the link path may be relatively small*

*compared to the attenuation caused by wet antennas, i.e., the wet antennas could contribute to some of the overestimations."* (for the case of rainfall overestimation over short-link paths); and *"The 1-min rainfall intensities from the 3 disdrometers from the region of São Paulo are also employed to estimate the value of $\alpha$ used to convert the minimum and maximum rainfall intensities from the HU CML to mean 15-min intensities. The found value, 0.30, is close to the default one in RAINLINK, 0.33, based on Dutch data and used in this study. This confirms the usefulness of the default value of $\alpha$ for application in a subtropical climate."* (for the case of the suitability of RAINLINK for subtropical climates). *Please find in our reply to comment "P8L14-18" of reviewer #3 details with regard to the insertion of this new text in the updated version of the manuscript.*

• Only being able to derive meaningful results for 5 out of 250 CMLs indicates that either the methods used by the authors are lacking or the technique of using CML data for rainfall estimation in general is less promising than expected.

*R/. We have now substantially improved the derivation of meaningful results. By means of implementing a gauge validation, and a modification in the RAINLINK code to derive rainfall intensities from minimum received power levels only, we now present meaningful results for a maximum of 116 CMLs. The analysis for 116 CMLs corresponds to a case in which we compared CML-derived rainfall against gauges up to a distance of 9 km in the vicinity of CMLs. If the vicinity is reduced to a 1 km, the results are meaningful for 35 CMLs. Still, from those 35 CMLs, we deemed as best performing CMLs those for which the relative bias is within ±25%, and for which the coefficient of correlation is above or equal to 0.6.*

*Figure 6 of the updated version of the manuscript (please see figure below) shows the results (and metrics) of these improved analyses. Overall, the presented metrics may suggest a poor performance of the network and of the RAINLINK estimates. Nevertheless, and throughout the updated version of the manuscript, we demonstrate how this technique is very promising, despite all the inconveniences found in the datasets.*

[Figure]

• The fact that the majority of the CML data, the Ericsson data which only provides the minimum signal levels, cannot be used with the existing codebase of the authors (RAINLINK) should not be an excuse for not analysing it. Rather this calls for adjusting or extending the existing code.

*R/. We have now extended the functionality of RAINLINK to be able to retrieve rainfall intensities from minimum received powers only. Thus, we have added 91 ER CMLs that were not previously analyzed. We now even have one best performing CML that came from this dataset. In the updated version of the manuscript, we indicate how this was done: "The ER CMLs only provide minimum power levels. RAINLINK is designed to retrieve rain rates from minimum and maximum power levels. Thus, in order for RAINLINK to compute mean rainfall estimates only from minimum power levels, two steps extra are required: 1) in the input file(s) for RAINLINK, the column with maximum power levels has to receive the values of the column with minimum power levels; 2) the mean path-averaged rainfall intensity, i.e. the output from RAINLINK, is now a maximum rainfall intensity and needs to be multiplied by a conversion factor to obtain the actual mean intensity…". Please find in the replies to your comments the complete updated text (and its placement in the manuscript).*

• The final analysis is based only on short or very short CMLs, but the authors do not state if they applied a wet antenna correction method, even though they note themselves that the effect of wet antenna can strongly impact shorter CMLs. This makes all the reasoning about biases arbitrary.

*R/. We do apply a fixed wet antenna attenuation correction as described in Overeem et al. (2016a), using the default value of 2.3 dB. Thus, in the last paragraph of sub-section "2.3 Rainfall Retrieval Algorithm", the sentence "4) rainfall retrievals -- once attenuation estimates are obtained from the difference between RSL and the reference signal level, 15-min average rainfall intensities are computed from a weighted average of minimum and maximum rainfall intensities obtained by the (inverse) power-law of Eq. (1);" was rephrased as **"4) rainfall retrievals -- once attenuation estimates are obtained from the difference between RSL and the reference signal level, a fixed wet antenna attenuation correction is applied (2.3 dB), and subsequently 15-min average rainfall intensities are computed from a weighted average of minimum and maximum rainfall intensities obtained by the (inverse) power law of Eq. (1);"**. Please see our remarks to your very first bullet (above) with regard to the effect of wet antenna on short CMLs.*

• The authors state that gauge records can also be unreliable, nevertheless they use low correlation with gauge records as indicator to neglect CML data.

*R/. As suggested in our reply to the second bullet of the reviewer, we have now implemented a gauge validation procedure in order to validate gauges, and CML retrievals more consistently. The validation procedure is as follows: 1) For every gauge (152 in total) the closest two gauges were selected for comparison; 2) For the entire period, 30-min rainfall pairs (dry periods included) were evaluated throughout the relative bias and the coefficient of correlation for both closest gauges; 3) If the metrics of at least one of the two closest gauges are within ±25% for the relative bias, and ≥ 0.6 for the correlation coefficient, the gauge under evaluation was deemed reliable.*

*We describe this procedure in the updated version of the manuscript by replacing the sentences "Stations located within 1 km distance from the evaluated link paths were selected. Hence, only 11 stations were used to evaluate CML rainfall estimates in Sao Paulo." (at the end of the second paragraph of sub-section "2.2 Data") by **"A gauge validation procedure was necessary due to availability issues and doubts about the quality of the rainfall observations from the CEMADEN gauge network. The validation procedure is as follows: 1) For every gauge (152 in total) the closest two gauges were selected for comparison; 2) For the entire period, 30-min rainfall pairs (dry periods included) were evaluated through the relative bias and the coefficient of correlation for both closest gauges; 3) If the metrics of***

*at least one of the two closest gauges are within ±25% for the relative bias, and ≥0.6 for the*
*correlation coefficient, the gauge under evaluation was deemed reliable. This selection results in 96*
*valid gauges out of 152. Comparisons of city-averaged rainfall were carried out among data from valid*
*(96), and all (152) gauges, and all (145) CMLs (Fig. 4). For comparisons of individual path-averaged*
*estimates of CMLs against gauges, only gauges within 1 or 9 km from the evaluated link paths were*
*selected. For the 1-km case 35 CMLs were compared against 20 gauges, whereas for the 9-km case 116*
*CMLs were compared against 87 gauges."*

*Given that rain gauges are the only available source we could refer our CMLs rain retrievals to, we still*
*consider that high values of $r^2$ indicate that both types of observations contain a true rain signal (with*
*lower correlations, it is actually not possible to know whether the inaccurate rainfall estimate comes from*
*the CML or from the gauge measurements). We reflect on this in the updated version of our manuscript*
*in the new paragraph **"Figure 7 shows the performance of individual CMLs by plotting the values of CV***
***against $r^2$, based on CML-gauge pairs both above 0.0 mm (for the studied period). Many CMLs have***
***fairly high values of $r^2$. For instance, 43% of the CMLs have a value of $r^2$ larger than 0.5 (for CML-gauge***
***pairs within 9 km). Here, CML and gauge measurements are totally independent. Thus, it is very likely***
***that the high values of $r^2$ for a large minority of CMLs indicate that both types of observations contain***
***a true rain signal.",** which was added before the end of the sub-section "3.1 Evaluation of 30-min*
*Rainfall" (now sub-section "3.2 Evaluation of 30-min Rainfall").*

*Figure 7 (see below) is a new figure in the updated version of the manuscript.*

[Figure]

**Recommendations:**

• I recommend an extensive major revision, i.e. a real extension of the current analysis (see my points below)

*R/. As the reviewer can see from the updated version of the manuscript, we carried out a much more*
*comprehensive evaluation of the performance of CMLs in the city of São Paulo, implementing almost of*
*the suggestions from the reviewers.*

• Given the seemingly very heterogeneous quality of the raw data set (which is fine for an opportunistic sensing technique like the one used here), the scientific focus should in my opinion be to describe how to cope with this data quality issue.

*R/. To provide suggestions on how to cope with the data quality issue is rather difficult given the errors in*
*the metadata and the lack of a study confirming the quality of our reference rain gauge data. However,*
*the RAINLINK package also includes several quality control steps, and we now explicitly mention a couple*
*of recommendations on using link length and frequency information for link metadata quality control.*
*Such recommendations are made in the sub-section "2.2 Data" of the manuscript, as follows:*

- *In the first paragraph, "From the 66 HU CML, we selected 17 CML given their proximity to rain gauges (1 km or less)." was replaced by* **"Figure 1 shows the location of these CMLs. [new paragraph] Figure 2 shows the scatter plot of link frequency against link length for all CMLs. In Fig. 2 the CMLs with uncommon or dubious (dub) combinations of length and frequency are denoted by gray markers (grey paths in Fig. 1)."**.
- *At the end of the first paragraph, the sentences "Hence, we discarded 6 CML as dubious and did not consider them in our analyses, which reduced the number of CML to 11. Finally, from the 11 remaining CML, we only kept the 5 CML which showed clear rainfall signals as compared to nearby rain gauges, i.e. for which $r^2 \geq 0.7$. The other 6 CML practically showed no correlation with nearby gauges ($r^2 \sim 0.3$ for one CML-gauge pair, and $r^2 < 0.1$ for the other 5 CML-gauge pairs), due to malfunctioning gauges and/or CML data issues. Figure 2 shows the scatter plot of frequency against length for all HU CML." were replaced by* **"The group of markers in the left bottom corner of Fig. 2 is also considered as dubious. Nevertheless, some CMLs around 7 GHz, having link lengths above 10 km, could still be realistic. We decided to only use the group of CMLs with path lengths shorter than 20 km and microwave frequencies above 15 GHz. Hence, 91 ER CMLs (40 link paths) and 54 HU CMLs (55 link paths) are left for the analyses, i.e., 145 CMLs in total (95 link paths)."**.
- *Footnote 4 (previously and wrongly in Pag. 5) "We received CML data from a third party. It was not possible to verify the topology of the network, shown in Fig. 1 on-site, which we suspect not always to be accurate given the orientation of the long links." was rephrased as* **"We received CML data from a third party. It was not possible to verify on-site the topology of the network shown in Fig. 1, which we suspect not to be accurate given the orientation of the long links.",** *and now it is inserted on Pag. 16 (where it should have been placed).*

*The updated Figure 2 is:*

[Figure]

• The constraint to neglect CMLs which are further than 1 km away from a rain gauge should be weakened. One can argue about what a "reasonable" threshold distance for comparing two rainfall measurements is. But, 1 km is really very strict, in particular, since the CMLs integrate over hundreds of meters or several kilometers anyway. The increased distance between CML and gauge will add additional uncertainty for sure, but when I look at the presented results and the relative biases from Table 1, having more data for the analysis seems to be more important than absolute accuracy of rain rates and/or rainfall sums.

*R/. We agree with the reviewer that the 1-km constraint removes a large part of the links. On the other hand, we would like to limit the influence of sampling errors on the analyses. In order to meet both of these requirements we now present a comparative analysis for two CML-gauge distances (1 and 9 km), and a global analysis of rainfall time series for all CMLs against average gauge accumulations.*

*For the comparative analysis of two CML-gauge distances (threshold distance) we decided to keep the one of 1 km, and use an alternative one of 9 km. This latter is based on the de-correlation distance (9.1 km) for 30-min rainfall in the city of São Paulo (value obtained from the gauge validation procedure). For each of these two distances (1 and 9 km), we also carried out two types of analyses: 1) where all possible paired rainfall depths were used; and 2) where only paired rainfall depths above 0.0 mm was used, i.e., to only account for significant/rainy events. Thus, in the updated version of the manuscript paragraphs 4 and 5 of sub-section "3.1 Evaluation of 30-min Rainfall" (now sub-section "3.2 Evaluation of 30-min Rainfall") were removed from the manuscript, and the following two paragraphs were inserted instead:*

**"Figure 6 shows an overall assessment of the CML performance to retrieve 30-min rainfall depths (over the studied period). Scatter density plots are for CML-gauge pairs within 1 km (top panels, a and b) and within 9 km (bottom panels, c and d). The left column (panels a and c) is for all CML-gauge pairs, whereas the right column (panels b and d) only includes rainy intervals, i.e., CML-gauge pairs where both rainfall depths are above 0.0 mm. The rainfall estimates for CML-gauge pairs within 1 km are somewhat better than the ones for 9 km in terms of $r^2$ and CV, but the relative bias of the latter is smaller than that of the former. If all CML-gauge pairs are used, on average CMLs underestimate rainfall by 23-29%, with high values for CV and low values for $r^2$. Assuming that the gauges provide reliable measurements, this performance indicates that the applied wet-dry classification could be sub-optimal. Perhaps a sensitivity analysis of the threshold values in the wet-dry classification could improve this classification. If only rainy intervals are used, i.e., CML-gauge pairs both above 0.0 mm, these lead to a strong reduction in the value of CV, a decrease in the $r^2$, and a much smaller relative bias. [new paragraph] A reason for the large discrepancies among the statistics of the scatter density plots (Fig. 6) could be the fact that only minimum (and also maximum for HU CMLs) RSL data is used to compute 15-min rainfall intensities, i.e., a limited temporal sampling. Rios Gaona et al. (2015) compare CML (actual) and gauge-adjusted (simulated) path-average rainfall depths for a 12-day dataset from the Netherlands, based on rainfall pairs for which at least one depth exceeds 0.1 mm. The most prominent difference is their much higher value for $r^2$ (0.437), which was found for 15-min rainfall. Hence, the sampling strategy is not necessarily the main reason for the low values of $r^2$. Given the erroneous metadata found in the CML dataset (Sec. 2.2), which led to discarding CMLs with dubious combinations of path length and frequency, there could be errors in the metadata from selected CMLs too, i.e, wrong location of one of the antennas or wrong frequency. In addition, although a basic assessment of gauge quality has been performed, even records from gauges classified as valid could still contain measurement errors.".**

*For the global analysis of cumulative rainfall series for all selected CMLs and gauges, averaged over the city of São Paulo, a new sub-section ("3.1 City-average Rainfall", right at the beginning of section "3 Results and Discussion") was added to the manuscript. This new sub-section focuses on the city-average performance of gauges and CMLs. The new paragraph reads as follows:* **"For each dataset we compute the cumulative city-average rainfall for the studied period (Fig. 4). According to the reference, i.e., the 96 valid gauges, the cumulative rainfall depth in this ~3-month period is ~600 mm. The differences in cumulative rainfall depths between the valid and all (152) rain gauges are small. Such a small difference suggests that the gauge dataset is reliable. For the "PreProcessed" CML dataset no wet-dry classification and no outlier filter are applied. This contributes to cumulative rainfall depths being roughly twice as large as the gauge-based ones. Moreover, the dynamics do not often correspond with that of the gauges, for instance around 1 December 2014. For the "OutFiltered" dataset of 145 CMLs,**

*which includes a wet-dry classification and outlier filter, a much better correspondence is found. The dynamics of the cumulative series agree reasonably well, and an overall underestimation is found, ~200 mm at the end of the period, albeit much smaller than the difference between the "PreProcessed" dataset and the reference. The separate performance of the HU and ER CMLs shows that the HU dataset performs quite well with some overestimation, whereas the ER dataset gives a huge underestimation, despite roughly capturing the rainfall dynamics."*.

*Also, the last paragraph of sub-section "2.4 Error and Uncertainty Metrics" "The metrics were systematically computed on 30-min paired rainfall depths, both above 0 mm (to only account for significant rainfall events), and for which their equivalent 15-min minimum received powers (i.e., "min PRx ..." in Fig. 4) were larger than -90 dB. 30-min aggregation was necessary given the temporal resolutions of the datasets, i.e., 10 min for gauge and 15 min for link-retrieved data." was rephrased as* **"The metrics were systematically computed on 30-min paired rainfall depths, using either all rainfall pairs or only pairs where both CML and gauge depths are above 0.0 mm. The latter to account only for significant rainfall events. 30-min aggregation was necessary given the temporal resolutions of the datasets, i.e., 10 min for gauge and 15 min for CML-retrieved data."**.

• The Ericsson data should be included, i.e. RAINLINK should be extended to be able to process this data, or other code should be written or reused.

*R/. The ER CMLs were included in the current analyses, and the methodology for RAINLINK to retrieve rainfall from minimum received powers only is clearly described in the revised version of the manuscript. Hence, the following two paragraphs were inserted at the end of sub-section "2.3 Rainfall Retrieval Algorithm" to provide a background on the retrieval of rainfall depths (from RAINLINK) for CML-measurements of only minimum power levels:* **"The ER CMLs only provide minimum power levels. RAINLINK is designed to retrieve rain rates from minimum and maximum power levels. Thus, in order for RAINLINK to compute mean rainfall estimates only from minimum power levels, two steps extra are required: 1) in the input file(s) for RAINLINK, the column with maximum power levels has to receive the values of the column with minimum power levels; 2) the mean path-averaged rainfall intensity, i.e. the output from RAINLINK, is now a maximum rainfall intensity and needs to be multiplied by a conversion factor to obtain the actual mean intensity. This conversion factor needs to be determined by means of a calibration dataset. Here, we use the 1-min rainfall intensities from the three disdrometers from the region of São Paulo to obtain an estimate of such a conversion factor. For each 15-min interval, the minimum rainfall intensity is selected from the lowest intensity of the 15 1-min intensities. 0.38 was found as the conversion factor, by comparing this minimum rainfall intensity against the mean 15-min rainfall intensity from the same disdrometers. ER-CML maximum rainfall intensities are then multiplied by this factor to obtain (actual) mean rainfall intensities. [new paragraph] The 1-min rainfall intensities from the three disdrometers from the region of São Paulo are also employed to estimate the value of $\alpha$ used to convert the minimum and maximum rainfall intensities from the HU CMLs to mean 15-min intensities. The found value, 0.30, is close to the default one in RAINLINK, 0.33, based on Dutch data and used in this study. This confirms the usefulness of the default value of $\alpha$ for application in a subtropical climate."**.

**Other major comments and questions:**

Page 4, line 22: What were the actual lengths and frequencies of the "long" CMLs? If the transmit power is high enough or large antennas are used, "uncommon" combination are possible. From Fig 1. some of the very long CMLs look strange indeed, though.

*R/. Please see the updated Figure 2 (please see the figure above in reply to the second recommendation of the reviewer), where frequencies against link lengths are shown for all possible CMLs in the revised*

*dataset (both HU and ER). In this figure there are several CMLs with frequencies close to 0. Personal communication with network design engineers from a telecommunication company in the Netherlands confirms that the discarded microwave frequency - path length combinations should be erroneous. Having a larger transmit power to compensate for longer path lengths is generally not used because of the greatly increased probability of interference with other systems in the same band.*

Page 6, line 13: A 50 km radius to look for CMLs with jointly decreasing power levels seems a bit large, in particular since, as the authors write in section 2.1 and 3.1, there is a lot of convective spatially very variable rainfall in the study region. Hence, is this radius of 50km too big? And how sensitive are the RAINLINK processing results on this threshold?

*R/. From the gauge validation procedure we found that the de-correlation distance of 30-min rainfall for the São Paulo are is 9.1 km. The figure below (not shown in the updated version of the manuscript) is a histogram of the distances at which the paired gauge evaluations comply with the thresholds of a relative bias within ±25% and a coefficient of correlation above 0.6. From this figure one can see that almost all of the distribution is within a 'radius' of 10 km (9.1 km being the arithmetic average). Hence, for the re-analysis presented in the revised version of the manuscript we modify the RAINLINK radius parameter to 9 km.*

[Figure]

*For the revised version of the manuscript, we use the "OutFiltered" RAINLINK-approach, i.e., the approach including wet-dry classification and outlier filter.*

Page 8, line 7: Limiting the analysis to CML-gauge pairs were both show a rainfall depth above 0 mm, neglects the validation of the challenging step of detecting rain events in the CML time series, which, to my understanding, is the first step in RAINLINK. Wrong detections, i.e. missed rain events or artificially generated rain, may considerably add bias to the accumulations. Hence, this effect should be included in the validation or added in a separate validation.

*R/. We agree with the reviewer that this is indeed an important aspect of CML rainfall retrieval. We have included such analyses in the revised version of the manuscript. Please see the first part in reply to the third recommendation of the reviewer, in which we explain in detail how these analyses were carried out. Please see the reply to the second "main concern" of the reviewer in which the support figure (new figure in the updated version of the manuscript) of the analyses of rainy events and all events with dry spells is presented.*

Page 8, line 31ff: Given that this is the result for 1 out of 250 CMLs, I would recommend

not to draw that optimistic conclusions based on the current state of the analysis.

*R/. For the updated version of the manuscript, we have now included the ER dataset, we have carried out more consistent and comprehensive analyses, such as city-average rainfall and wet and wet-dry spells, and we have even tripled the amount of "outstanding" results we have gathered for the first version. Such good and promising findings were updated accordingly in the conclusion section.*

*Thus, in the section "Summary, Conclusions and Recommendations", the second paragraph "30-min rainfall estimates from CML were evaluated against rainfall measurements from the nearest rain gauge for the period from 20 October 2014 to 9 January 2015. We focused our analyses on the 5 CML for which $r^2>0.7$. Three out of five CML gave good results in terms of CV. One CML also had a low relative bias. Subsequently, the quality of rainfall estimates from these 5 CML was also evaluated in terms of cumulative rainfall from 272 events. The good results indicate that RAINLINK can successfully be applied to CML data from a subtropical climate, even though most parameters have been optimized for the temperate climate of the Netherlands." was rephrased as "30-min rainfall estimates from CMLs were evaluated against rainfall measurements from rain gauges for the period from 20 October 2014 to 9 January 2015. Despite the mixed results, the potential of CML technology for rainfall estimation in subtropical climates is confirmed. Especially, given the rainfall dynamics captured by the city-average rainfall (Fig. 4), the good performance of some individual CMLs (Fig. 5), and a high correlation for a large minority of CMLs (Fig. 7). This gives an indication that the RAINLINK package is suitable to retrieve rainfall via CML data from a subtropical climate, even though many of its parameters have not been optimized for such a climate. Since biases propagate in hydrological model predictions, given the low relative bias found for rainy periods (Fig. 6), CML rainfall estimates could be considered as an alternative input in hydrological models.".*

Fig 1: As it is mentioned in the text, the very long CMLs indeed look strange since they do not even end on one of the summit of the mountains in the north and north-east. Wouldn't it be possible to check via GoogleMaps satellite images if there is a relay or cell phone tower there? It would be nice to have a more solid basis for neglecting these CMLs. At least give more details in the text. Maybe it would also be good to show two or three maps, one with all CMLs, one with "reasonable" CMLs and in addition only the CMLs used for analysis (which hopefully will be much more in the next revision of the manuscript: : :).

*R/. Checking the locations of the antennas of links on e.g. Google Maps could indeed be a valuable addition. However, the effort of manually checking antenna locations is not feasible for the large dataset we are dealing with here. It is also important to realize that antennas that were previously used for other links could have been re-used without having changed the location metadata in the database (a likely error; personal communication with representatives from a cellular communication company in the Netherlands). This means that there are likely still antennas at that location, but the specific antenna will have moved. Hence, checking for the presence of antennas on Google Maps will likely not yield the necessary information.*

*Please also see our reply to the second recommendation of the reviewer, in which we gave more precise arguments on how to discard dubious or erroneous link paths.*

*We have managed as well to implement in only one figure the suggestions of the reviewer concerning the display of used and discarded CMLs for the respective analysis. The figure below is the updated Figure 1 in the revised version of the manuscript.*

[Figure]

**Technical and minor comments** (this is a uncomplete list, since I assume that the manuscript will considerably change with the next iteration):

Fig 2: I only see 4 crosses not 5 as indicated in the caption. Also the red circles and red crosses seem not to add up to 11. Maybe overplotting is an issue here. If yes, this should be mentioned. Furthermore, no CMLs longer than 8 km are shown, even though the caption states that all HU CMLs are plotted, for which, according to Fig 1., some are definitely longer than 8 km.

*R/. The scatter plot the reviewer refers to has been updated in the revised version of the manuscript (please see our reply to the second recommendation of the reviewer, in which the updated figure is presented). The axes have been extended to show the characteristics of all possible CMLs. Now, Figure 2 is completely consistent with Figure 1 (figure immediately above).*

Fig 5: The two yellowish colors are hard to distinguish. Anyway, if colors are different, markers could maybe be the same to make the graph easier to read. Or even better, have separate scatter plots for the CMLs, or at least for selected ones, if the number of CMLs increases with an extended analysis.

*R/. This figure (and section) has been removed from the manuscript. It does not appear in the revised version of the manuscript.*

Table 1 and Table 2: The relative biases are exactly the same in both tables. As far as I understood, Table 2 is based only on a subset of the rain events from Table 1. Hence, I assume there is something wrong with either Table 1 or Table 2.

*R/. Given that we now carried out different analyses (focused on the suggestions of all the reviewers), such tables are not needed anymore. Thus, these tables have been removed from the manuscript. They do not appear in the revised version of the manuscript.*

Table 1 and Table 2: Is CML 12 and 13 along the same path, but just the two directions?
*R/. Exactly. Nevertheless, as mentioned in the previous reply these tables do not appear in the revised version of the manuscript.*

**Anonymous Referee #2**

The paper proposes to analyze an important topic : possible use of CMLs data for quantitative rainfall estimation in one of the largest city under tropical climate, i.e Sao Paulo. As reminded by the authors the area is prone to intense rainfall, leading to flash floods and other natural hazards such as land slides. The authors and various other groups have already demonstrated the potential of the CMLs based method under a range of climate and weather situations ( from widespread systems in the Netherland to intense convection in Africa, through Mediterranean areas and even mountainous regions). This new data set in Brazil is an opportunity to test the CMLs method in a more challenging context then in previous studies: the quality of the CMLs data set is not homogenous, the validation network is sparse. The authors seems to have partially avoided this challenge by focusing only on a very limited subsample of the data set (where and when it works: : :.); unfortunately this also limits the scientific impact of the study and its interest as a demonstrator of CMLs potential for hydro-meteorological monitoring over Sao Paulo: : :.
Given the existing literature on the CMLs topic and the extensive data set available here, the present study should be taken a step further and provide a more robust and extensive analysis of the available data set, including issues related to data quality, sparse GV and data format variation among CML providers..
*R/. We thank the reviewer for the constructive review. We have now extended the analyses by including all the ER CMLs. We have implemented the capabilities of RAINLINK to retrieve rainfall depths from only minimum received powers (which is the case of the ER dataset). We have also carried out analyses of rainfall retrievals not only for gauges within 1 km in the vicinity of selected CMLs but also for vicinities up to 9 km. The data quality is still difficult to investigate in more detail because of erroneous metadata or the lack of information regarding the quality of the rain gauge network used as a reference. Nevertheless, we have implemented a basic quality check on the gauge data to remove 'malfunctioning' gauges.  Please see our replies to reviewer #1, especially those for "main concern" # 5 (fifth bullet), and recommendations # 2, 3, and 4 (second, third, and fourth as the reviewer actually does not provide numbers).*

A major limitation of the paper in its present form is that conclusions are drawn from a very limited subset of the available data set : only a few links (5 out of a possible total of above 200 ) are exploited and for theses links the analysis is restricted to time steps where both the link and the nearby gauge detect rainfall. Doing so the authors miss a major issue : capability of the method to detect rain and not generate false alarms, and so over the whole network.
*R/. As mentioned in the previous reply, in the revised version of the paper we have now carried out analyses on the whole dataset, i.e., HU + ER CMLs. We have also analyzed CML retrievals from wet, and dry-and-wet spells, to account for effectiveness of estimating zero rain.
Please see our replies to reviewer #1, especially those of "main concern" # 2, and third recommendation (first part), in which we addressed specifically these issues.*

This a major forthcoming of an otherwise very well written paper, which also provides a good review of the state of the art in CMLs based rainfall estimation. I can only encourage the authors to take the necessary time to submit a improved version of their work and take the analysis a step further.

Detailed major/minor recommendations :
-One important feature of sub-tropical rainfall is the occurrence of intense (and possibly extreme) rainfall rates associated with convective cells. This is very important for some of the applications the authors put forward in their introduction . No information is provided on the actual rain rate distribution (at the 30 minutes time step for instance) observed over the study period in Sao Paulo by the gauges and how well ( or not) the CML method retrieves it. The global statistics provided in Table 2 and 3 do not inform us on the performance of the Rainlink/CML data according to rain rate classes . This is an important question, for hydrological applications for instance.
*R/. We agree with the reviewer that this is indeed a relevant question. However, we feel that this is outside of the scope of the present paper and a topic for future research.*
*This recommendation (jointly with others) was taken into account in the revised version of the manuscript, as follows:*
*In the previous to the last paragraph the sentences **"We did not evaluate the performance of CML-RAINLINK retrievals based on rain rate classes. Nevertheless, this evaluation is highly encouraged as it would shed some light on the suitability of CMLs for hydrological applications, for instance. [new paragraph]"** were inserted after the period in "... paths. We...".*
*Also, the sentences **"Note that the value of $\alpha$, estimated from local 1-minute disdrometer rainfall intensities, was close to the default value from RAINLINK. Especially the value of A_a and the threshold values for the wet-dry classification and the outlier filter should be investigated."** were inserted after the period in "... regions. Missing...".*
*In its last sentence (paragraph previous to the last one) "This shows that accurate metadata, such as link coordinates for instance, are essential." was rephrased as **"This shows that accurate metadata, such as link coordinates for instance, are essential, as well as the feedback about obtained CML and reference datasets."**.*

-Selection of the time steps and 'events'. The authors should provide statistics covering the whole analysis period and not solely on a selected number of 30' times steps. Time step where one OR the other sensor detected rain should be included and a contingency table provided. The definition of 'events' , as presented in Fig 5 is not clear. Does it include some non rainy time steps or is it based on the same selection as the 30 ' (both CMLS and gauge > 0)? Daily statistics would be useful and would allow comparisons with other studies : : :.
*R/. We decided not to include daily statistics in the updated version of the manuscript. Nevertheless, we now compute statistics for the entire period of study (~3 months). For this updated version of the manuscript, we removed our analyses on 'rain events'.*
*Please see our replies to the two reviews (previous to "Detailed major/minor recommendations:") of the reviewer.*

-The authors mention wet antenna as a possible source of bias : this should be explored further - The order of magnitude of wet antenna attenuation is known, is it compatible with the observed bias ?
*R/. We already apply a fixed wet antenna attenuation correction of 2.3 dB (please see our reply to the fourth "main concern" of reviewer #1). This is just an average value. For a given rainfall event, the wet antenna attenuation may differ a lot since one, two or no antennas can become wet. Moreover, the amount of attenuation can also depend on rainfall intensity, and biases can also be caused by other*

*phenomena. Hence, it is rather difficult to assess whether remaining biases are caused by wet antenna attenuation. At least part of the wet antenna attenuation has been corrected for.*

-CMLS data selection : the authors should extend the analysis to other CMLs links even if they keep the present 5 links to illustrate the best case– This is important to asses the actual potential of the method in a context representative of reality. Given that the analysis is carried out at the 30' and 'event' time step, 1 km maximum distance from the gauge seems very severe.

*R/. As also noted in previous replies to the reviewer, we have extended our analyses by adding the ER dataset. We have also carried out analyses within 1 and 9 km in the vicinities of selected CMLs (116 in total).*
*Please see our replies to reviewer #1, especially those concerning the third recommendation (first part) and the fourth recommendation.*

The conclusions should be revised once a truly extensive assessment has been done on this data set. I am looking forward to see a revised version that will investigate further this rich data set acquired in Brazil !
*R/. As noted in all the previous replies to reviewers #1 and #2, the conclusions of the revised version of the manuscript have been updated accordingly.*
*Also, the fourth paragraph of the section "4 Summary, Conclusions and Recommendations" was removed.*

**Anonymous Referee #3**

**General comments:**
The manuscript Rainfall retrieval with commercial microwave links in Sao Paulo Brazil aims to evaluate potential of commercial microwave links (CMLs) as rainfall sensors in the subtropical climate. The authors collect data from several microwave links, process them using RAINLINK R package and compare them to rain gauges operated by CAMADEN. Moreover, disdrometer observations are used to estimate parameters of attenuation-rainfall power-law model and these parameters are compared to those from ITU recommendations and from Dutch case studies.
Although the topic of CML rainfall retrieval in subtropical climate is relevant and the presented dataset is valuable the study has several major drawbacks: i) the authors select for evaluation only well performing CMLs. This is a reasonable approach if the selection procedure is independent of a reference rainfall data. However, this is not the case, as one of the selection criteria is correlation of CMLs to the reference rain gauges ii) the results are presented and discussed very briefly without sufficient attempt to investigate the causes of good/bad performance of particular CMLs. The influence of drop size distribution to the attenuation-rainfall model is analyzed in more detail, however, this effect can explain only small fraction of total errors. Especially spatial representativeness of reference rain gauge data should be more properly analyzed to avoid interpreting discrepancy between path-integrated CML rainfall observation and point RG rainfall observation as an inaccuracy of a CML iii) The conclusions are not sufficiently supported by the data. The authors claim that CMLs are very promising source of rainfall data only based one very good and two relatively well performing CMLs. Also the suitability of RIANLINK package for CML processing in subtropical regions is not proofed. The data rather indicate that constant WAA correction used in the RAINLINK package is inappropriate for CMLs.

Given the above mentioned shortcomings the reviewer does not recommend the manuscript for a publication, however, encourage the authors to improve the data analysis, rewrite especially the results, discussion and conclusion sections and resubmit the manuscript. Some suggestions for revisions are given in the specific comments bellow.

**Specific comments:**
The reviewer suggests changing the structure of the manuscript: i) moving the descriptions of the evaluation procedure (event definition) from the Results and Discussion section to the Data and Methods section, ii) considering moving the results of DSD analysis from the Rainfall retrieval algorithm section.

*R/. We feel that the description of the evaluation procedure is an important part of the Results and Discussions section, and that moving this part would not increase the readability of the paper. The same holds for the DSD analyses and the Rainfall Retrieval Algorithm section. We will therefore keep the structure of the paper as it was.*

**P4L26:** Is the threshold value r2 _ 0.7 chosen arbitrary? Why not 0.5 or 0.9? In any case, the selection of CMLs for evaluation based on reference data does not enable to evaluate potential of CMLs without having reference rainfall. This is one of the major drawbacks of the whole analysis. Moreover, it might be valuable keeping the bad performing CMLs in the analysis and identify the causes of the bad performance.

*R/. We agree with the reviewer that the value of 0.7 is indeed arbitrary. We have now implemented a basic quality control to the gauge data to remove erroneous gauges, so that this issue will be less important. We have also removed the constraint that $r^2$ should be above 0.7.*
*Please see our replies to reviewers #1 and #2, especially that of reviewer #1 in his/her third recommendation (first part).*
*It was really difficult to identify the causes of bad performance as we did not get additional information about the link network.*

**P5L5:** Given the CML paths lengths from several hundreds of meters up to several km the criterion of 1 km distance from link path seems to be too strict and not always reasonable. E.g. for CML 14 it might be more representative to use average of two RGs even though the second RG is several km far away. In any case, the reviewer suggests presenting at least some basic analysis of RG correlation and set the criterion based on this analysis. Such analysis would also support the results and enable to distinguish between discrepancy of path and point measured rainfall and between errors due to inaccuracy of CMLs.

*R/. As noted in previous replies to the reviewer, we have extended our analyses by adding the ER dataset. We have also carried out analyses within 1 and 9 km in the vicinities of selected CMLs (116 in total). Please see our replies to reviewer #1, especially those concerning the third recommendation (first part) and the fourth recommendation.*

**P6L21:** The section describes rather in detail generally well known performance metrics, however does not provide complete information about evaluation procedure. E.g. it should be explained here how the event based evaluation is performed (metrics are calculated for each event and then averaged as presented in Tab 2?).

*R/. As also noted in previous replies, we decided not to include daily statistics in the updated version of the manuscript. Nevertheless, we now compute statistics for the entire period of study (~3 months). For this updated version of the manuscript, we removed our analyses on 'rain events'. Tables 1 and 2 were also removed.*

**P7L15:** why -90 dB and not some other value?

*R/. Because now we present analyses based on the "OutFiltered" RAINLINK approach, we use the default RAINLINK value, i.e., -32.5 dB km$^{-1}$ h$^{-1}$.*

**P7L17:** Both overall evaluation and event based evaluation is presented here. This is very good idea, as one could learn e.g. during which types of events CMLs perform well. However, at the end the event based results are presented in overall statistics (Tab. 2) except results presented in the Fig. 5. It might be very interesting to see how stable the CML performance is (e.g. in terms of variance of the metrics). This could be presented as boxplots or scatter plots of metrics, similarly as on Fig. 5. This would also enable more proper discussion of the results with potentially answering to questions like these: Do CMLs perform better during strong rainfalls than light rainfalls?
Do they better reproduce rainfall temporal dynamics (r2) during light or heavy rainfalls?

*R/. We agree with the reviewer that this is indeed interesting to know. However, we feel that this is beyond the scope of this paper. Nevertheless, in the previous to the last paragraph, after the period in "... paths. We...", we have introduced the following recommendation:* **"We did not evaluate the performance of CML-RAINLINK retrievals based on rain rate classes. Nevertheless, this evaluation is highly encouraged as it would shed some light on the suitability of CMLs for hydrological applications, for instance. [new paragraph]".**
*For more details, please see our reply to the first comment in "Detailed major/minor recommendations" of reviewer #2.*

**P8L6-L11:** The event definition might be rather in the method section

*R/. We removed the analyses on 'rain events'. Please see our reply to the earlier comment about this.*

**P8L14-18:** It seems that shorter CMLs are substantially more biased than longer CMLs. This indicates that the bias arises from wet antenna attenuation. Thus, RAINLINK's representation of baseline (constant) seems not working very well.

*R/. As noted in our reply to the first "main concern" of reviewer #1, we were able to identify the three best performing CMLs. The particularity of these three best CMLs is that they are shorter than 1.7 km, where representativeness errors play a smaller role. We also found overestimations in CML estimates (Figs. 4 and 6 of the updated version of the manuscript). Such overestimations may be related to the fact that rain-induced attenuation along the link path may be relatively small compared to the attenuation caused by wet antennas, i.e., the wet antennas could contribute to some of the overestimations.*
*The above discussion is inserted in the third paragraph of sub-section "3.1 Evaluation of 30-min Rainfall" (now sub-section "3.2 Evaluation of 30-min Rainfall"), where the sentences "The results of Fig. 4 are obtained for the longest link (5.3km), where representativeness errors could play a larger role. The worst results are found for the shortest links (<1.0km). This may be related to the fact that rain-induced attenuation along the link path may be relatively small compared to the attenuation caused by wet antennas, i.e., the wet antennas may explain the large overestimation found for CML 13 and 12 (see Tables 1 and 2)." were rephrased as* **"The results of Fig. 5 are obtained for short links (< 1.7 km), where representativeness errors will play a smaller role. Overestimations by CMLs may be related to the fact that rain-induced attenuation along the link path may be relatively small compared to the attenuation caused by wet antennas, i.e., the wet antennas could contribute to some of the overestimations.".**

**P8LL35 – P9L2:** The performance was clearly very good only for one CML whereas the other experience relatively high bias. This is not really proving the good performance of RAINLINK in subtropical regions.

*R/. We have now demonstrated the suitability of RAINLINK for a subtropical climate to a greater extent. Please see our reply to the first "main concern" of reviewer #1.*

**P9L18-20 and P10L4-6:** Only three CMLs out of 17 resp. 11 were identified (based on reference rainfall) as well performing. The suitability of RAINLINK for processing such data should be, therefore, discussed more critically. Similarly, the authors claim that the potential of CMLs would be great if the data and metadata are properly stored. This is unfortunately not happening in the reality as demonstrated by the presented results.

Thus, use of CMLs for subtropical regions is still rather big challenge. The dataset presented in this paper might, however, contribute to coping with this challenge. Thus, the reviewer highly encourages the authors to invest more work into its analysis and resubmit the improved manuscript.

*R/. As noted in our replies to reviewers #1 and #2, we have substantially improved our manuscript.*

**Fig.1:** CMLs selected for the analysis are really tiny in the figure. Maybe cropping and resizing the figure would help (long CMLs aiming to the north-west could be cropped as they are not used for the analysis).

*R/. Figure 1 has been updated in the revised version of the manuscript. Please see our reply to the last comment of "Other major comments and questions" from reviewer #1.*

**Tab. 2:** It seems to be there is no distinctive difference in the effect of DSD when evaluated over the whole dataset (tab. 1) and event based. It might be, therefore, reasonable to present here only results for fitted DSD (i.e. best performing a, b parameters) and instead one value (Mean of a metric?) present e.g. mean and standard deviation of a metric.

*R/. As noted in our previous replies, Tables 1 and 2 have been removed from the manuscript.*

**OTHER COMMENTS.**
The following are some other changes that have been implemented in the revised version of the manuscript:

*The amount of CMLs and its distribution was updated accordingly throughout the whole manuscript.*

*In the abstract, "Results were found to be promising and encouraging, especially for short links, for which high correlations (>0.9) and low biases (~30% and lower) were obtained." was rephrased as* **"Results were found to be promising and encouraging when it comes to capturing the city-average rainfall dynamics. Mixed results were obtained for individual CML estimates, which may be related to erroneous metadata."**.

**"Uijlenhoet et al. (2018) give a non-expert summary of the history, theory, challenges, and opportunities toward continental-scale rainfall monitoring via CMLs of cellular communication networks."** *was added as the last sentence at the end of the first paragraph of Pag. 3 (submitted version).*

*"Because our CML retrieval algorithm RAINLINK (Sec. 2.3) only retrieves rain rates from minimum and maximum power levels, we discarded the ER CML. Due to issues in the log-file of the attenuation measurements, it was only possible to correctly and unequivocally assign power levels to 66 HU CMLs and*

147 ER CMLs." was replaced by **"The ER CMLs are assumed to have constant transmitted power levels."** in the first paragraph of sub-section "2.2 Data".

The sentence "For the remaining 5 CML evaluated here, the mean difference between 15-min transmitted power levels is ~0.0 dB, with a maximum of 0.5 dB (for the 81 days considered)." was removed from the manuscript.

Sub-section "3.2 Evaluation of Event Rainfall Accumulations" was erased from the manuscript and replaced by the new sub-section "3.2 Evaluation of 30-min Rainfall".

The paragraph **"For the studied period, we evaluate the quality of 30-min path-averaged rainfall estimates from individual CMLs against gauges by: 1) time series from rainfall events for the three best performing CMLs; 2) scatter density plots based on data from all CMLs; and 3) metrics for each CML."** was added at the beginning of sub-section "3.1 Evaluation of 30-min Rainfall" (now sub-section "3.2 Evaluation of 30-min Rainfall").

At the end of the first paragraph of sub-section "3.1 Evaluation of 30-min Rainfall" (now sub-section "3.2 Evaluation of 30-min Rainfall"), the sentence "The figure presents the two longest rainfall events for CML 14." was rephrased as **"The figure shows that these three CMLs capture reasonably well two of the rainiest events of the studied period."**.

The first half of the last paragraph of sub-section "3.1 Evaluation of 30-min Rainfall" (now sub-section "3.2 Evaluation of 30-min Rainfall") "The results for different R-k relations are quite similar, indicating that differences in DSD climatologies play a smaller role. For CML 12 and 13 the relative bias becomes less severe for the R-k relation derived from São Paulo data." was rephrased as **"The presented results are based on the R-k relation derived from São Paulo data, which is representative for the local rainfall climatology. The results (not shown here) for the different R-k relations are quite similar (Sec. 2.3), which indicates that differences in DSD climatologies play a smaller role."**.

At the end of sub-section "2.2 Data", the sentences "The DSD recorded by the Parsivels were corrected by the method of Raupach and Berne (2015a, b). We use updated correction factors trained from French disdrometer data. Due to instrumental, climatic, and location differences, these correction factors are taken as approximations." were removed from the manuscript.

Summary
15/04/2018 20:44:34

Differences exist between documents.

**New Document:**
04_paper_v08d3
22 pages (1.93 MB)
15/04/2018 20:44:11
Used to display results.

**Old Document:**
04_paper_v07d2
21 pages (1.82 MB)
15/04/2018 20:44:11

Get started: first change is on page 1.

No pages were deleted

**How to read this report**

**Highlight** indicates a change.
 indicates deleted content.
🔺 indicates pages were changed.
↔ indicates pages were moved.

[revised manuscript text omitted]

---

## Author Response (AR2)

\_\_\_\_\_

Suggestions for revision or reasons for rejection (will be published if the paper is accepted for final publication)

As already mentioned in the first review (and by the other reviewers) assessing the potential of CMLs for rain retrieval in a new region (here Brasil) could be a great addition to the work that has already been provided on this topic by several groups in Europe, Israel and Africa. Compared to other studies on the same subject and in the sub-tropics the data set available here is much richer (both in terms of available links and gauges ? with also a nearby disdrometer) and of great potential interest for demonstrating the advantages and limits of CMLs based rainfall measurement.

The present work is unfortunately far from delivering the full potential of the available data set.

Instead of the authors ?encouraging future work? on the data set and listing in their conclusion some of the many things that ?could be done?, one feels like encouraging the authors themselves to take the current analysis a step further in order to take better advantage of the data set and draw some convincing conclusions to ?shed some light on the suitability of CMLs? . This would be useful for the CML and hydrometeorology community.

R/. We thank the reviewer for recognizing the potential of our study, but we respectfully disagree with the reviewer's assessment of the first revised version of our manuscript. In our opinion, we performed extensive and quantitative analyses, which have been significantly expanded in our revised manuscript. In the Conclusions section, we 'encourage future work' "on sensitivity analyses focused on the optimization of RAINLINK parameters to improve the accuracy of rainfall estimates in subtropical regions". With this, we consider that future work should be focussed on the fine-tuning of RAINLINK parameters with the purpose of achieving a higher degree of accuracy in rainfall estimates (from RAINLINK) in other locations around the world. In our updated analyses, despite the specific issue of lacking CML data and metadata for the São Paulo metropolitan area, we managed to reveal a clear rain signal and provide an honest assessment of the quality of CML rainfall estimates.

The authors have added some additional work compared to their discussion paper but most of the reviewers? comments are far from being accounted for in this new version.

R/. We also regret that the reviewer thinks that we did not take into account most of the reviewers' suggestions. In the 17-page rebuttal we submitted, we implemented on average 80% of all the reviewers'

suggestions, including the majority of the suggestions to perform additional analyses. We implemented these major changes in our first revised version, showing that we substantially revised our manuscript:

- We modified the RAINLINK algorithm in order to also retrieve rainfall depths from measurements of only minimum received power and we derived an optimal conversion factor to retrieve mean rain rates using disdrometer data from Brazil. Therefore, we were able to also systematically analyse 147 ER (Ericsson) CMLs.

- Although this was not specifically suggested, we added an independent gauge validation procedure in order to only select trustworthy gauges for the validation of CML rainfall estimates.

- Correlation with gauge records is not used anymore to remove or keep CMLs in the dataset.

- Scatter density plots of half-hourly CML rainfall depths vs. gauge rainfall depths over the entire 2.5-month period were added. Comparisons are not only presented for maximum CML-gauge distances of 1 km, but also 9 km.

- Although this was not suggested, we included cumulative time series of 30-min rainfall averaged over the city of São Paulo, Brazil, for CMLs and gauges over a 2.5-month period.

- Scatter plots of the performance of individual CMLs against gauges (coefficient of variation against coefficient of determination) for maximum CML-gauge distances of 1 and 9 km were included.

- Instead of a 50-km radius, a 9-km radius was used in the wet-dry classification. In contrast to the first version of our manuscript, all results in the first revised version have been processed employing the wet-dry classification.

- Comparisons are not limited anymore to CML-gauge pairs were both show a rainfall depth above 0 mm, but also include all rainfall pairs, i.e. zeros included.

- All the figures were updated.

The main problem is that the results of the CML-gauge comparison as presented (for instance Fig 6) are mostly showing that the method, as applied here, fails to reproduce the 30? rainfall time series satisfactorily. And because most of the links are in practice unusable and unreliable for Quantitative precipitation estimate, the use of a CML network to obtain high-resolution rainfields and quantify intense rainfall (for urban hydrology, landsliding risk detection etc.) as proposed in the introduction is a very bad idea!. The CML technique may be used to have a very rough (and quite biased) estimate of average rainfall over a city (cf Fig 3).

Some links (3?) behave well at least for a few events (Fig 4), however the ?well-functionning? links are detected a posteriori, thanks to the gauges, so not very usable in practice.

R/. We do recognize that mixed, not purely bad, results are obtained for the CMLs in São Paulo, but we strongly believe that papers with mixed results should also be published. We provide an honest piece of work clearly describing which CMLs have been filtered and why. Apparently, results are not always as good as found in other studies. We also notice that (part of the) metadata is erroneous, hence serving the community by pointing to specific problems which can be encountered in CML rainfall estimation. In our paper we do not claim that all

these CMLs are useful to obtain high-resolution rainfields and quantify intense rainfall, but earlier studies do reveal this potential (e.g. Overeem et al., 2016). In our opinion, the dynamics of city-average rainfall in Figure 4 are reproduced fairly accurately. We have shown in (new) Figure 4 the cumulative average time series for all the 145/213 CMLs over the city of São Paulo (for 2.5 months of data). In that figure, one can see that for almost half of the studied period (i.e., 1.5 month continuously) the CML-series is in agreement with the global-/city-average obtained from gauge data. Even after a discrepancy for one intense rainfall event, the dynamics of the CML time series are remarkably similar to those of the gauge data. In addition, we also find a high correlation for a large minority of individual CMLs (Figure 7). Hence, we disagree that only 3 CMLs behave well. We also disagree with the reviewer's comment to question the validity and practicality of our method on the premise of an "a priori" rain gauge evaluation. Modern techniques of rainfall retrieval, such as radar and satellites, still use rain gauges for validation and calibration of their products, and thus their improvement is also "detected a posteriori".

The authors have dismissed the suggestions by several reviewers of the discussion paper, to understand better why so many links are in disagreement with the gauges and decompose and quantify the problem step by step (miss/false-detection ? why ? ).

R/. We have attempted to provide some explanations as to why the performance of several links is very bad (e.g., p.9, lines 6-14). Having said that, we think that more elaborate analyses of why several links do not seem to reproduce rainfall values is outside the scope of this paper. The uncertainty analysis was from the beginning not the aim of our current manuscript. Moreover, it will be very difficult to systematically explore these quality issues, because these are probably related to erronous metadata. Unfortunately, several attempts have not led to any help from people knowledgable on the CML datasets in São Paulo. Such involvement seems necessary to take this a step further. Finally, we developed a gauge validation procedure, which makes it more likely that disagreement between link and gauge estimates is due to the links.

One of Rainlink?s step is to compare the consistency of a link with its neighbors ? Couldn?t this feature be further exploited to detect the consistency among links and understand the problem - before comparison with gauges ?

Here the assessment is performed using the 30? time series, would results be better -in terms of detection at least- at the daily time step ?

R/. The use of the comparison of a link with its neighbors would yield information about the consistency of the occurrence of attenuation among links. If there is a lack of consistency this would mean that either the rain is highly variable or that there is a problem with a link. We disagree with the reviewer that this could provide much help for finding the causes of the bad link rainfall estimates, especially because we have an independent reference (gauges). A link wrongly identifying rainy periods can be caused by many different phenomena, e.g. dew formation on the antennas, and reflection or refraction of the beam. A link removed by the outlier filter

points to a malfunctioning link, but does not provide a clear reason. Note that the wet-dry classification and the outlier filter are performed each 15-min time step, but already incorporate information from the previous 24 h. For instance, the outlier filter actually discards a time interval of a link for which the cumulative difference between its specific attenuation (based on uncorrected minimum received power) and that of the surrounding links (i.e. within a radius of 9 km) over the previous 24 h becomes lower than the outlier filter threshold. A comparison of CML and gauge rainfall estimates on a daily time step generally improves results and could be interesting. Note that we do already provide city-average rainfall accumulations, showing a clear rain signal over a 2.5-month period, i.e. over longer durations.

Other more minor point :

I also regret that the authors dismissed the suggestion made by more than one reviewers to provide not only global statistics but also statistics by rain classes, or at least quantifying if the links perform well in heavy or light precipitation. Once again, given that the authors put forward hydrology, floods, land-sliding as applications, and given the stress put in testing the method in sub-tropical climate with intense rainfall rather then in The Netherland, an analysis of the performance in heavy rainfall is very relevant.

R/. Although this could be an interesting analysis, we feel that this is outside of the scope of the present paper and a topic for future research. Note that Figure 6 provides scatter plots for the full range of rainfall depths over the entire 2.5-month period, hence already giving an indication of the performance as a function of rain classes.

Once AND IF a serions effort for improving the content is done, the english text will also need revising.

The presentation of the data, quality control, signal processing (outliers elimination etc.) and results still lack precision and clarity. (for instance still some confusion between R=ak^b and k=aR^b although already commented for in first review ) ?

Some of the processing choices or data filtering appear quite arbitrary and should be better argued for and their impact quantified.

See detailed comments below.

R/. We note that anonymous referee #2 stated in the review of the first version of our manuscript that "This a major forthcoming of an otherwise very well written paper, which also provides a good review of the state of the art in CMLs based rainfall estimation.". Nevertheless, we have checked the paper for spelling and grammar errors, and corrected these where applicable. We thank the reviewer for noting that on one occassion the k-R instead of R-k relationship was used. We changed this into R-k relationship in Eq. 1. We reply to the other points in response to the detailed comments below.

DETAILED COMMENTS :

Section 1 Intro - Introduction

p2 l4

?backscattring (i.e. reflectivity) is not the only way to measure rainfall with radar, polarimetric radar may propagation parameters such as specific differential phase shift for accurate measurement of heavy rainfall. As a matter of fact the city of São Paulo is equipped with (at least) one polarimetric radar. ?

R/. The sentence "The accuracy of rainfall estimates from radar depends on how well the measurements of backscattered power from hydrometeors are transformed into rain rates." was rephrased as "The accuracy of rainfall estimates from radar depends on how well measurements of received signal power from hydrometeors or specific differential phase shift are transformed into rain rates."

p3 l24 : Sahel may be semi arid but rainfall in this region also falls as intense events (cf many recent articles on floods and rainfall intensification in Sahel) and is associated with deep convection ? the sentence is misleading. What may oppose the region is the terrain and oceanic influence in SP while flat/continental environment in Sahel.

R/. We agree with the reviewer and changed the text accordingly into: "Doumounia et al. (2014) focused on a semi-arid, tropical climate. Our evaluation is one of the first which focuses on a humid subtropical climate."

Section 2 DATA

P4 I14 : do you mean that for ER you have only received and not transmitted power ? Please clarify as this in an important point for attenuation processing.

R/. The sentence "The ER CMLs are assumed to have constant transmitted power levels." was rephrased as "As indicated by the metadata (i.e., lack of information for the transmitted power), the ER CMLs are assumed to have constant transmitted power levels.".

P4 l18-19 : doubt on the length of links - couldn?t this information been checked on site ? from your brasilian partners ?

Some free internet resources as for instance the site

**http://telecocare.teleco.cl9.com.br/telebrasil/erbs/**

provides exact locations of RF antennas from all operators in Brasil ??.you may check some of the links displayed in Fig 1.

R/. We asked our Brazilian partner several times for more information on (meta)data, but this did, unfortunately, not yield additional information. We know the website the reviewer recommends. It will be hard to obtain additional information on the accuracy of link locations using this website. First of all, the website does not provide microwave link locations, but cell tower locations. I.e., it could only be verified whether a cell phone tower is present, but no information is provided on the connections between cell towers. In addition, our dataset is from 2014 and 2015, whereas the cellphone tower locations on the website probably provide the current locations. The TIM network seems very dense, and our dataset seems to contain only part of the data from their network. Hence, the usability of such a website is rather limited.

P4 I 28-29 : ?closest two gauges? ? Given the density of the network I assume it means very close (less than 1 km ? ) in absolute term. But please provide an indication of the max range considered here.

R/. "closest two gauges" exactly means "closest two gauges" regardless the distance that separate them. It could be as short as, e.g., 200m or as long as 50km. In our previous rebuttal we presented a histogram of the pairing distribution of rain gauges. We showed in that figure that ~80% of paired-gauges lay within 6 km. The sentence "For every gauge (152 in total) the closest two gauges were selected for comparison;" was rephrased as "For every gauge (152 in total) the closest two gauges were selected for comparison (note that ~80% of paired gauges lay within 6 km);".

P5 I 1 : what is the rationale for these threshold values (bias and r2) ? they seem arbitrary unless you explain why they were chosen.

R/. Naturally, the choice of threshold values is somewhat arbitrary. We consider an  $r2 \ge 0.6$  and  $rB \le +25\%$  as adequate enough for the analyses we carried out. These values imply a reasonable agreement between nearby gauges, while still leaving some room for differences due to spatial rainfall variability.

P5 ? L17 : CML operating frequencies range from 7 to 80 Ghz (at least) depending on regions, regulations, length etc? b is not equal to 1 for the whole range. Please be more specific.

R/. We replaced "In the frequencies at which CML commonly operate, the exponent in Eq. (1) is ~ 1.0." with "For the majority of frequencies at which CML commonly operate (~13-40 GHz), the exponent in Eq. (1) is close to unity (i.e. between 0.8 and 1.2)".

P5 l28 : here you use R=ak^b and in (1) k=aR^b. Please be careful ? these inconsistencies in k-R vs R-k were already pointed out in the first review.

R/. This has been solved, since we changed this into R=ak^b in Eq. (1).

**P6 1rst paragraph ? Figure 3**

?it is clear from the figure that there certainly are differences ? ? - Please provide a more quantitative assessment of these diffrences between the curves and between frequency ? and add on figure or provide in text the values of the a,b coefficients for comparison.

R/. We consider that the difference between frequencies are clearly provided by the color scale. In the same way, we do not consider that crowding the Fig. 3 with 18 values of coefficients a and b (or even adding a table) will contribute further to the discussion we already presented in this paragraph.

P6 paragraph 2 ? Rainlink algo description :

1) at which time scale is done the comparison with nearby links to assess dry/wet ? 15 minutes ? please clarify

R/. The sentence "1) wet-dry classification - a link is considered for non-zero rainfall retrievals if the received power jointly decreases with that of nearby links (9-km radius for this study);" was rephrased as "1) wet-dry classification - for each 15-min interval (RAINLINK's default), a link is considered for non-zero rainfall retrievals if the received power jointly decreases with that of nearby links (9-km radius for this study);".

3)outlier removal : what do you mean exactly by ?deviates too much? ? how do you accumulate specific attenuation over 24 hours ?

R/. We replaced "exclusion of links for which the specific attenuation (accumulated over 24 h) deviates too much from that of nearby links" with "exclusion of a time interval of a link for which the cumulative difference between its specific attenuation (based on uncorrected minimum received power) and that of the surrounding links (i.e. within a radius of 9 km) over the previous 24 h becomes lower than the outlier filter threshold (-32.5 dB km^-1 h)".

4) what is the rationale for the value 2.3 dB ? is it applied what ever the frequency of the links ?

R/. This value is taken from the calibration by Overeem et al. (2013). A detailed sensitivity analysis has been carried out by Overeem et al. (2016) showing that this value is appropriate for a 2.5-year dataset from the Netherlands, also for different seasons. The majority of CMLs in that study have a microwave frequency of 37-40 GHz, but also CMLs with lower frequencies, down to 13 GHz, are used. They state that "the insensitivity of the parameter values (Aa and alpha) to season, and hence, to rainfall type, holds a promise for applying the optimal values to link data from other climates". Moreover, employing 12 days of data they report fairly similar values for Aa for frequency classes of 10-20, 20-30 and 30-40 GHz. Hence, we decided to use the value 2.3 dB for all CMLs in our study, irrespective of their microwave frequency.

р7

**Section 3 ? results**

P8 I1 ? ?Such a small difference (in 3 month accumulations) suggests that the gauge data set is reliable? ? As well know by the authors agreement in terms of bias over a 3 month period does not mean he series is reliable as a validation data set used at the 30? time step. Please be more serious in the assessment of the gauge data set.

R/. We agree that this small difference between validated and unvalidated gauges only reveals an agreement in terms of bias. In the gauge validation we excluded 56 gauges due to their relatively low correlation and/or high relative bias with respect to nearby gauges. The claim that all gauge data are reliable is therefore too strong. Hence, we decided to remove this sentence.

P8 l12 ? ?three best performing CMLs? - How exactly was that assessed ? are these the best performing CMLs over the whole period ? best performing in terms of which cretiria ?

R/. The sentence "1) time series from rainfall events for the three best performing CMLs;" was rephrased as "1) time series from rainfall events for the three best performing CMLs (i.e., CMLs for which r2>=0.6 and rB<= - +25% against their respective closest gauge);".

P8l17 ? the fig shows that the se three CMLs capture reasonably well tow of the rainiest events? ? NO it doesn because on 1 CML is shown for the 2 events !!

Fig 5 = why aren?t the 3 links shown for the 2 events ? this would be much informative.

The lines for the 2 ?upscaled series? should be made more visible ? as this is what we actually want to compare.

R/. No, this is not true. Figure 5 shows four panels, three CMLs (052, 041, 135, as indicated in the figure) and two events. Showing three CMLs for either of the two events is not possible due to availability issues in the CML

data. Hence, we decided to keep the figure as it stands. Moreover, the figure is meant as an illustration, whereas other figures contain more global statistics on the whole CML dataset.

P8 L25 : I very much doubt that this is has any effect on the short links presented in Fig 4 and for 30? average.

I suggest using the dense available gauge network to check what the spatial decorrelation of the 30? average rainfall actually is for the SP.

R/. We thank the reviewer for noting this. As previously stated in reply to the reviewer's comment "P4 I 28-29", in our previous rebuttal we presented a histogram of the paired distribution of rain gauges. We also showed in that figure that the decorrelation distance for 30-min rainfall in São Paulo is ~8 km. Hence, the sentence "This is, on average, not the case here though as we found a decorrelation distance of ~9 km for 30-min rainfall in the city of São Paulo (not shown here)." was added at the end of the paragraph the reviewer refers to.

P8 l29 : the authors should make the effort to quantify this point (relative impact of wet antenna vs rain attenuation along the way) using the present or/and their other data sets?. Is there evidence that this bias in attenuation/rainfall is more present just after than just before the storm ?

R/. We remind the reviewer that the aim of this manuscript was not the quantification of such sources of uncertainty, and especially not in such a detail, and especially not with the quality of the available metadata. Our mainly aim for this work was a very first, relatively straightforward, but quantitative application of the RAINLINK algorithm to CML-data from São Paulo (despite all its issues of availability and metadata). Moreover, studying differences in attenuation before and after a storm will be hard in case of minimum and maximum powers over 15-min intervals. The quantification of wet antenna attenuation should ideally be done in an experimental setting with frequent logging. See our recent work on an experiment in the Netherlands over a 2-km link path (van Leth et al., 2017) and other studies on this topic (e.g. Leijnse et al., 2008; Schleiss et al., 2013).

**CONCLUSION**

The conclusion will have to re-written once the necessary additional analysis, which is suggested but not performed by the authors, has been done.

R/. The conclusions were indeed re-written in accordance with all the implemented changes that we carried out for the (previous) revised version of our manuscript. Please see our reply including the list of major changes at the beginning of this rebuttal.

**/R. References**

Leijnse, H., Uijlenhoet, R., and Stricker, J. N. M.: Microwave link rainfall estimation: Effects of link length and frequency, temporal sampling, power resolution, and wet antenna attenuation, Adv. Water Resour., 31, 1481-1493, doi:10.1016/j.advwatres.2008.03.004, 2008.

Overeem, A., Leijnse, H., and Uijlenhoet, R.: Country-wide rainfall maps from cellular communication networks, P Natl Acad Sci Usa, 110, 2741?2745, https://doi.org/10.1073/pnas.1217961110, 2013.

Overeem, A., Leijnse, H., and Uijlenhoet, R.: Two and a half years of country-wide rainfall maps using radio links from commercial cellular telecommunication networks, Water Resour Res, 52, 8039?8065, https://doi.org/10.1002/2016WR019412, 2016.

Schleiss, M., Rieckermann, J., and Berne, A.: Quantification and Modeling of Wet-Antenna Attenuation for Commercial Microwave Links, IEEE Geosci. Remote Sens. Lett., 10, 1195-1199, doi:10.1109/LGRS.2012.2236074, 2013

van Leth, T. C., Overeem, A., Uijlenhoet, R., and Leijnse, H.: An urban microwave link rainfall measurement campaign, Atmos. Meas. Tech. Discuss., https://doi.org/10.5194/amt-2017-404, in review, 2017.